# Hydrodynamic Performance and Design Evolution of Wedge-Shaped Blocks for Dam Protection against Overtopping

**Francisco Javier Caballero** [1,2,*], **Miguel Ángel Toledo** [1], **Rafael Moran** [1,3] and **Javier San Mauro** [3]

1 SERPA Dam Safety Research Group, Department of Civil Engineering: Hydraulics, Energy and Environment, E.T.S. de Ingenieros de Caminos, Canales y Puertos, Universidad Politécnica de Madrid (UPM), Profesor Aranguren s/n, 28040 Madrid, Spain; miguelangel.toledo@upm.es (M.Á.T.); r.moran@upm.es (R.M.)
2 ACIS Innovation+Engineering S.L. (ACIS2in), Planeta Urano 13, P18 2°A, 28983 Parla, Madrid, Spain
3 Centre Internacional de Metodes Numerics en Enginyeria (CIMNE), Campus Norte UPC, Gran Capitán s/n, 08034 Barcelona, Spain; jsanmauro@cimne.upc.edu
* Correspondence: franciscojavier.caballero@upm.es; Tel.: +34-616-978-633

**Abstract:** Dam safety requirements have become stronger in recent years, highlighting, among other issues, the need to increase the discharge capacity of existing spillways and the protection of embankment dams against potential overtopping, which are particularly threatened by the hydrological consequences of climate change. The current economic situation requires solutions that ensure the safety of these infrastructures at an affordable cost. Wedge-shaped blocks (WSBs) are one of these solutions. A more detailed understanding of the performance of WSBs was the objective of this work and, based on this, the evolution of WSB design. An extensive empirical test program was performed, registering hydrodynamic pressures on the block faces and leakage through the joints between blocks and their air vents. A new WSB (named ACUÑA) with a different design of air vents was tested in comparison to Armorwedge™, which was used as a reference case. Moreover, the hydraulic behavior of the WSB was analyzed according to the saturation state of the granular drainage layer. The ACUÑA unit was designed with air vents in the upper part of the riser where the registered negative pressures were higher. Negative pressures were also measured at the base of the block when the granular drainage layer was not fully saturated. Finally, the beneficial effect of sealing some of the joints between blocks was quantified.

**Keywords:** wedge-shaped block; WSB; overtopping; dam protection; dam spillway; dam safety; ACUÑA





## 1. Introduction and Background

Wedge-shaped blocks (WSBs) are modular elements made of precast concrete. They are intended to prevent the erosion and scour in soils caused by water flowing at high velocity. Such blocks are considered one of the feasible alternatives to protect embankment dams against erosion caused by overtopping [1]. They are installed in overlapping rows on the downstream shell of embankment dams or levees. Typically, the WSBs are manufactured with high-strength concrete, although initially, steel-reinforced blocks were also used [2].

The former idea of protecting dams against overtopping by overlapping concrete blocks comes from Gordienko, from the Moscow Institute of Civil Engineering in the late 1960s [3]. Subsequently, new advances were carried out by Pravdivets [2,4], Bramley, May and Baker [3,5–9], Clopper [10], Slovensky [11], Gaston [12], Frizell [13,14], Thornton et al. [15] and Relvas and Pinheiro [16–20], among others [21]. From this knowledge, the first technical guide to build spillways using wedge-shaped blocks was published [3], and the Armorwedge™ patent was developed by the U.S. Bureau of Reclamation (US5544973A). Additionally, several dams in operation have been built with this technology since 2007. Barriga dam in Spain [22] (Figure 1 and Figure S1 in Supplementary Materials), Bruton

dam [23], Ogden dam [24,25] and Norton-Fitzwarren dam [26] were built in the United Kingdom, and Friendship Village auxiliary spillway was built in Missouri, USA [1].

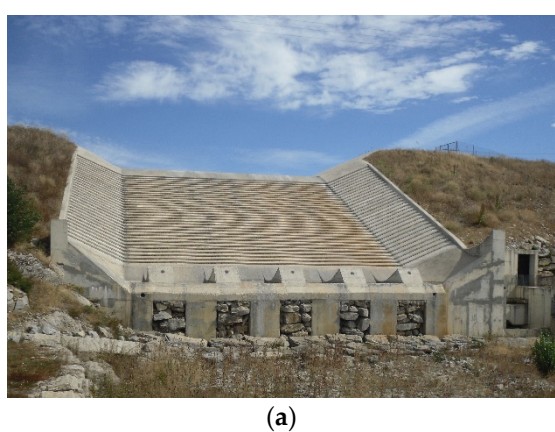 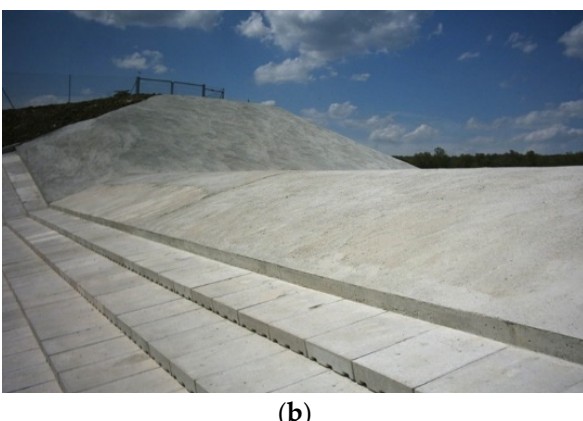

(**a**)                                      (**b**)

**Figure 1.** (**a**) The Barriga dam (Burgos, Spain). (**b**) The upper area detail of the WSB chute spillway of the Barriga dam.

The vents' surface and their position in the blocks are essential for the correct operation of WSB. A recent investigation in Spain, which is presented here, has focused on this subject. Earlier, in the 1990s, researchers from the University of Salford obtained the critical drainage surface in the block to achieve a satisfactory reduction of uplift pressure [3] and also performed a comparison between the drainage of the WSBs through slots located in the lower area of the overlapping zone between blocks and orifices located in the block tread [6,8]. In most of the experimental research, the setup of the tests was developed by placing the blocks over a drainage layer, which may become saturated at a certain inlet flow rate ($q$) value [5–9,11,17,19]. The uplift developed at the base of the blocks was a key aspect of their stability. As every block overlaps the ones located at the downstream row, the revetment works hydraulically as a stepped spillway. A fraction of the inlet flow leaks through the joints between adjacent blocks and inlets at the drainage layer under the blocks, usually formed of gravel material. The resulting seepage flow at such a drainage layer is critical for the stability of the block. [15,16]. The hydraulic stability of WSBs is mainly achieved due to three effects: the positive pressure of the main discharge flow, which impinges on the upper face of the next blocks downstream; the overlapping of the different rows; and the development of negative pressures on the block tread. Such stability is enhanced by the effect of the vents (also termed as "air vents" or "holes") transmitting the suction generated on the block tread towards the block base when the drainage layer is not saturated (Figure 2a). If the underlay is saturated, such suction may cause the return of a fraction of the drainage flow to the spillway chute, reducing the uplift pressure under the blocks (Figure 2b). In supplementary materials, some videos are included where this behavior can be appreciated (folders "01_No_saturation_*d2*" and "02_Suction"). WSBs have proven to be highly stable even in very unfavorable conditions [3].

There are references for this protection system in dams under 18 m high, with a maximum unit discharge of 3.9 m$^2$ s$^{-1}$ [1]. Nonetheless, good and likely better behavior should be expected for a greater velocity due to the positive effect of suction.

Since 2011, new research efforts have aimed to complement the theoretical and practical knowledge of WSB technology [27–31]. Such work aims to deepen our knowledge on an alternative technology in order to improve dam safety, which is particularly threatened by the hydrological consequences of climate change [32–39]. Specifically, the main goal of the research was to increase the cost efficiency of the discharge capacity of existing spillways and the protection of earth dams against potential overtopping. One of the results obtained is the development of a new design of WSB, ACUÑA (patented on 8 May 2017 ES2595852), which aims to improve the behavior of pre-existing blocks that have air vents in the lower part of the riser (i.e., Armorwedge™). The new design aims to improve the transmission of negative pressures and, therefore, the stability of the block. The new design also aims to

achieve other additional construction improvement objectives for the implementation of the blocks.

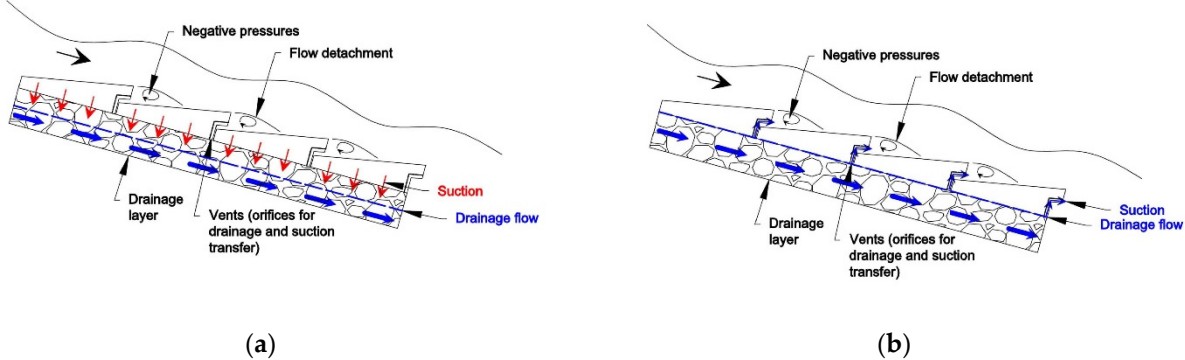

**Figure 2.** The hydraulic performance of a WSB spillway: (**a**) the drainage layer not saturated and (**b**) the drainage layer saturated.

## 2. Materials and Methods

### 2.1. Experimental Facility

The laboratory flume (located at CEDEX Hydraulics Lab, Madrid, Spain) includes a 0.50 m wide steel and methacrylate chute. The vertical slope is 2H:1V, and the maximum vertical drop is 4.7 m (Figure 3a). The side walls of the chute are 0.85 m high in the direction normal to the bottom. The maximum inlet flow rate ($q$) available is 0.24 m$^2$ s$^{-1}$. The water supply is carried out through water pumps from the lower tank to the inlet tank with a maximum elevation of 6 m from the lab floor and a horizontal area of 2.5 × 2.5 m$^2$. The tank has a 0.5 m wide and a 0.75 m high lateral opening, which connects to the chute by a 1.5-m-long horizontal inlet (Figure 3b). The WSBs tested in each trial are laid over the bottom of the chute and placed in 47 horizontal rows (Figure 3c). At the downstream end of the chute, there is a rectangular stilling basin that dissipates the energy of the flow. The measurement of the discharge is performed at a rectangular thin-plate weir at the end of the stilling basin before the flow is conveyed back to the lower tank. Next to the chute, on one of the side walls (in row 32), there is an outlet pipe of the seepage discharge in order to measure the drainage flow under the rows of WSBs by means of a triangular thin-plate weir. Additional photographs, schemes and videos of the experimental facility have been included in supplementary materials (Figure S2 in Supplementary Materials and 'folder 00_Experimental_set_up').

The instrumentation setup is able to measure the water level, the inlet flow ($q$) and drainage flow ($q_d$) discharge and the pressures on different positions of the blocks. Two methacrylate WSBs were designed as measuring blocks built with methacrylate sheets (Figure 4a). On such blocks, pressure sensors were installed on the block faces at different positions (Figure 4b). The methacrylate measuring blocks were installed on rows 5, 10, 15, 25, 30 and 35 (colored black in Figure 3c) to achieve measurements at different positions along the longitudinal profile of the chute. The measuring devices can be grouped as follows:

1.  Measuring devices for water levels and discharge of skimming and seepage flows:

    *   Electromagnetic flowmeter to measure the pumped flow rate.
    *   Triangular thin-plate weir to measure the flow that leaks through the open joints between adjacent blocks and seeps through the granular layer.
    *   Electromagnetic limnimeters (4) for measuring the water level at the following points: the inlet tank, the upstream end of the chute, the abovementioned triangular thin-plate weir and the rectangular thin-plate weir at the end of the stilling basin.

2. Pressure measurement system to register the water pressures at several points on the block tread, base and the riser step of the WSBs [27], formed by:

- A set of 12 Messtech submersible XA-700 pressure transducers connected to measuring tubes installed on one of the measuring blocks.
- A pressure gauge (Scanivalve DSA3207 Corp. model) with 12 sockets measuring tubes installed on a second measuring block.

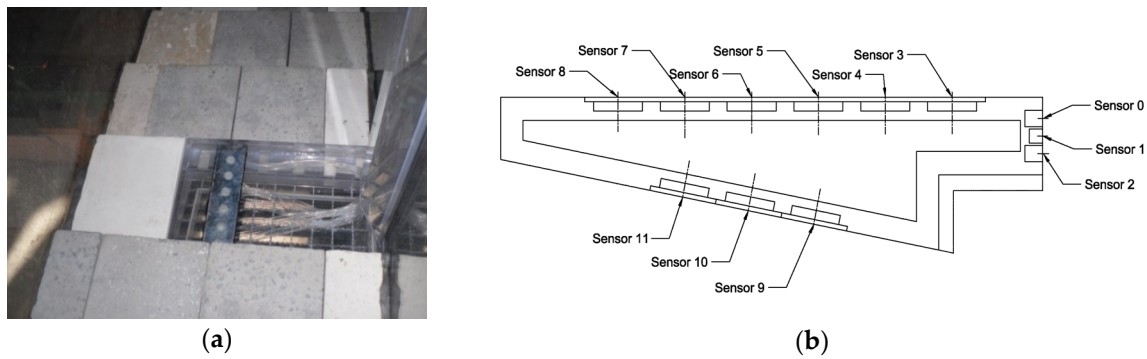

(**a**)                                                (**b**)

(**c**)

**Figure 3.** The experimental setup: (**a**) a photo of the testing facility, (**b**) scheme and (**c**) a pthe instrumented rows.

(**a**)                                                (**b**)

**Figure 4.** The measuring methacrylate WSB (**a**) and a block section with the locations of pressure sensors indicated (**b**) [28].

The data acquisition was performed at a time interval of 5 min per test for each inlet flow rate ($q$). Such acquisition was conducted using the Messtech submersible XA-700 pressure transducers (National Instruments) data collection equipment (cDAQ), which allows one to obtain 30 items of data per second and channel, and an Ethernet connection system by Scanivalve DSA3207 Corp., which allows one to obtain 70 items of data per second and channel. Furthermore, a conventional video camera was used for recording the tests.

### 2.2. Flow Test Characterization

#### 2.2.1. Flow Regimes

The type of flow regime (nappe, transition or skimming) determines the pressure pattern of the stepped chutes. For example, with a transition flow, the suction on the base of the block is lower than with skimming flow [18]. Several authors [2,13,18] relate the slope of the pseudo-bottom (i.e., the straight line connecting the step edges) to the ratio between the flume critical depth ($h_c$) and the height of the block riser ($h_s$), $h'$ (Figure 5, Table 1). Some of them [18] were used here to predict the flow regimes for each tested unit discharge ($q$).

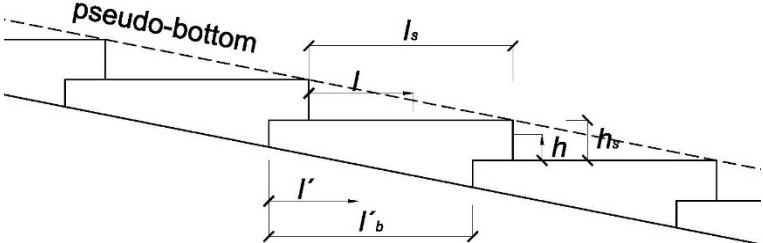

**Figure 5.** The geometric parameters of the block.

**Table 1.** Values of $h'$ and $q$ used in the tests.

| $h'$ ($h_c\ h_s^{-1}$) | 4.51 | 3.99 | 3.44 | 2.84 | 2.52 | 2.17 | 1.37 |
|---|---|---|---|---|---|---|---|
| $q$ (m$^2$ s$^{-1}$) | 0.24 | 0.20 | 0.16 | 0.12 | 0.10 | 0.08 | 0.04 |

In order to compare results with other authors, dimensionless distances were used; $l$ is the distance measured from the upstream end of the exposed tread (position of the pressure measurements on each sensor), and $l_s$ is the total exposed length of the upper surface of the block. At the base of the block, $l'$ is the distance measured from the upstream end of the base (position of the pressure measurements on each sensor), and $l'_b$ is the total length of the base of the block. The tread length ($l$) refers to the unit length ($l_s$) (Figure 5) to obtain the dimensionless parameter ($l\ l_s^{-1}$).

Figure 6 shows that every unit discharge used during the tests corresponded to skimming flow according to every author, except for Chamani-Ratjaratnam [40] and André [41]. According to the criteria proposed by Chamani-Ratjaratnam, the two lower discharges of the present study correspond to the transition flow, and all others correspond to the skimming flow. According to André, only the lower discharge corresponds to the transition flow.

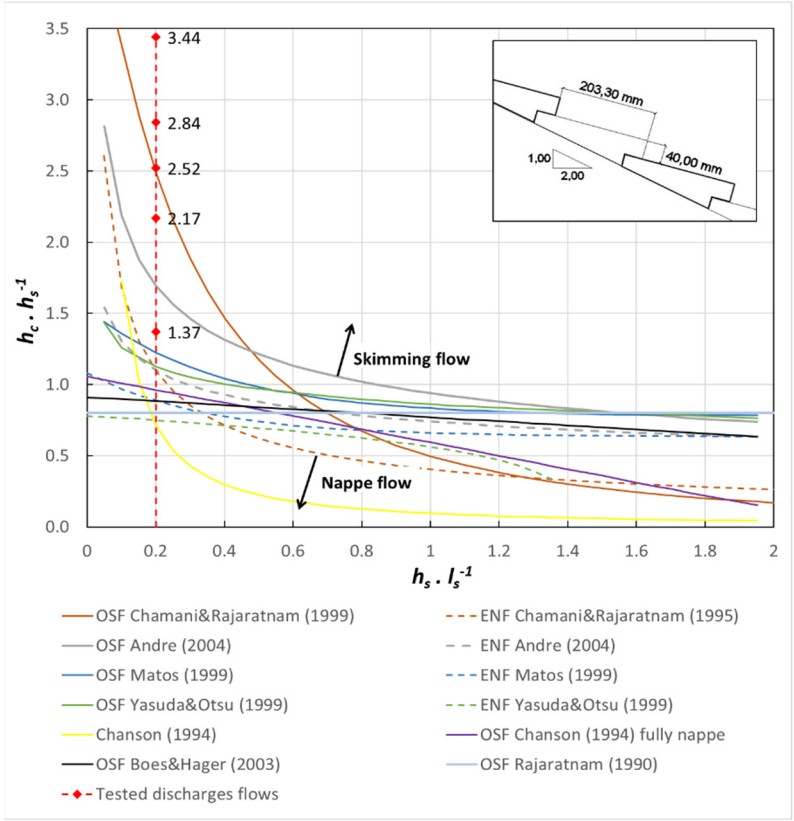

**Figure 6.** Flow regime (skimming, transition and nappe flow) for the tested discharges according to the criteria of [40–45].

### 2.2.2. Inception Point

The inception point is located at the position where the boundary layer intersects with the free surface of the flow. Such location is important to establish the upstream limit of air entrainment through the flow surface.

The location of the inception point (Table 2) was determined by visual observation according to the criteria of Mateos and Elviro [46]; i.e., the location was established where a permanent presence of air bubbles was observed for all the tested discharges. The observed location was compared with the results of the empirical formulas proposed by Relvas [18], Chanson [43,47] and Matos [44]. Results are included in Figure S3 of the Supplementary Materials.

**Table 2.** The inception point location of tested flows according to different authors.

| Flow Rate $q$ (m² s⁻¹) | $H'$ ($h_c\,h_s{}^{-1}$) | Row Number | | Distance of the Inception Point $L_i$ | | |
|---|---|---|---|---|---|---|
| | | Observed | Observed Interval | Relvas [18] | Chanson [43] | Matos [44] |
| 0.04 | 1.37 | 3 | 673–880 | 779 | 684 | 617 |
| 0.08 | 2.17 | 4–5 | 1035–1293 | 1283 | 1121 | 1035 |
| 0.10 | 2.52 | 5–6 | 1190–1500 | 1507 | 1314 | 1223 |
| 0.12 | 2.84 | 6 | 1293–1500 | 1718 | 1496 | 1401 |
| 0.16 | 3.44 | 6–7 | 1293–1707 | 2114 | 1837 | 1737 |
| 0.20 | 3.99 | 7–8 | 1500–1914 | 2482 | 2154 | 2053 |
| 0.24 | 4.51 | 9–10 | 1914–2328 | 2831 | 2453 | 2352 |

The results obtained in the research carried out by Gaston [12] presented a good agreement with the formulation of Relvas [18] for slopes of the chute of 2.5H:1V and 2H:1V.

However, the values of the roughness Froude number ($F^*$, Equation (1)) used by Gaston were much higher than the ones used by Relvas and ourselves in the present study.

$$F^* = q / \sqrt{g \cdot sin\theta \cdot (h_s \cdot cos\theta)^3} \tag{1}$$

defined in terms of $h_s$ and $\theta$, where $\theta$ is the angle formed by the flume with the horizontal.

Some previous formulations, such as that of Matos [44], do not take into account the influence of the slope of the channel, and the results are relatively close (but higher) to those obtained here by visual observation for the highest flow rates (from 0.16 m$^2$ s$^{-1}$).

### 2.2.3. Uniform Flow Area

If the channel is long enough, the uniform regime is reached. Then, the amount of entrained and aspirated air would be equal. Flow can be considered stable in this area. The upstream limit of the uniform flow area (Table 3) was estimated by means of empirical formulas [3,18,48]. Results are also included in Figure S4 of the Supplementary Materials.

**Table 3.** The uniform flow depth location (distance and row) of tested flows according to different authors.

| Flow Rate $q$ (m$^2$ s$^{-1}$) | $h'$ ($h_c$ $h_s$$^{-1}$) | Uniform Flow Depth Location (m) | | | | | |
| --- | --- | --- | --- | --- | --- | --- | --- |
| | | CIRIA Guide Hewlett et al. [3] | | Boes and Minor [48] | | Relvas [18] | |
| | | Distance (m) | Row | Distance (m) | Row | Distance (m) | Row |
| 0.04 | 1.37 | 1.09 | 5 | 1.83 | 8 | 1.56 | 7 |
| 0.08 | 2.17 | 1.73 | 8 | 2.91 | 13 | 2.57 | 12 |
| 0.10 | 2.52 | 2.01 | 9 | 3.38 | 16 | 3.01 | 14 |
| 0.12 | 2.84 | 2.27 | 10 | 3.81 | 18 | 3.44 | 16 |
| 0.16 | 3.44 | 2.75 | 13 | 4.62 | 22 | 4.23 | 20 |
| 0.20 | 3.99 | 3.20 | 15 | 5.36 | 25 | 4.96 | 23 |
| 0.24 | 4.51 | 3.61 | 17 | 6.05 | 28 | 5.66 | 27 |

### 2.3. Testing Program

The tests were carried out for different inlet flow rates ($q$) from a minimum value of 0.04 to a maximum of 0.24 m$^2$ s$^{-1}$ (Table 1). As has been noted, the goal was to define a new, more stable block. The experimental methodology was divided into several phases. Firstly, we carried out a comprehensive analysis of the performance of the Armorwedge$^{TM}$ block (Figure 7a), which was used as a reference as it has been successfully applied in actual cases thus far [1,22]. Specifically, the research focused on the effect of both the hydrodynamic pressures and the leakage flow through the joints among the blocks and the aeration vents. In addition to this, the uplift pressure generated by the seepage flow through the drainage layer was also measured. Initially, the Armorwedge$^{TM}$ block was tested without a granular support layer in free drainage conditions. Thus, every leakage flow was conducted separately over the bottom of the chute and below the metallic grid which supported the blocks (Figure 8a). In several trial tests, the rapid stabilization of the hydrodynamic pressures on the different sensors was verified, an aspect that was achieved in a few seconds. All the tests for the determination of hydrodynamic pressures were carried out at least twice to corroborate the results obtained.

Based on the obtained results and additional numerical research [27,30], the proposed WSB, ACUÑA, was designed for testing in the phases of the test program described below (Table 4 and Figure 8b). Air vents were located in the upper part of the riser, where the greatest negative pressure was achieved according to previous research [9,11,17], and the experimental results are discussed below.

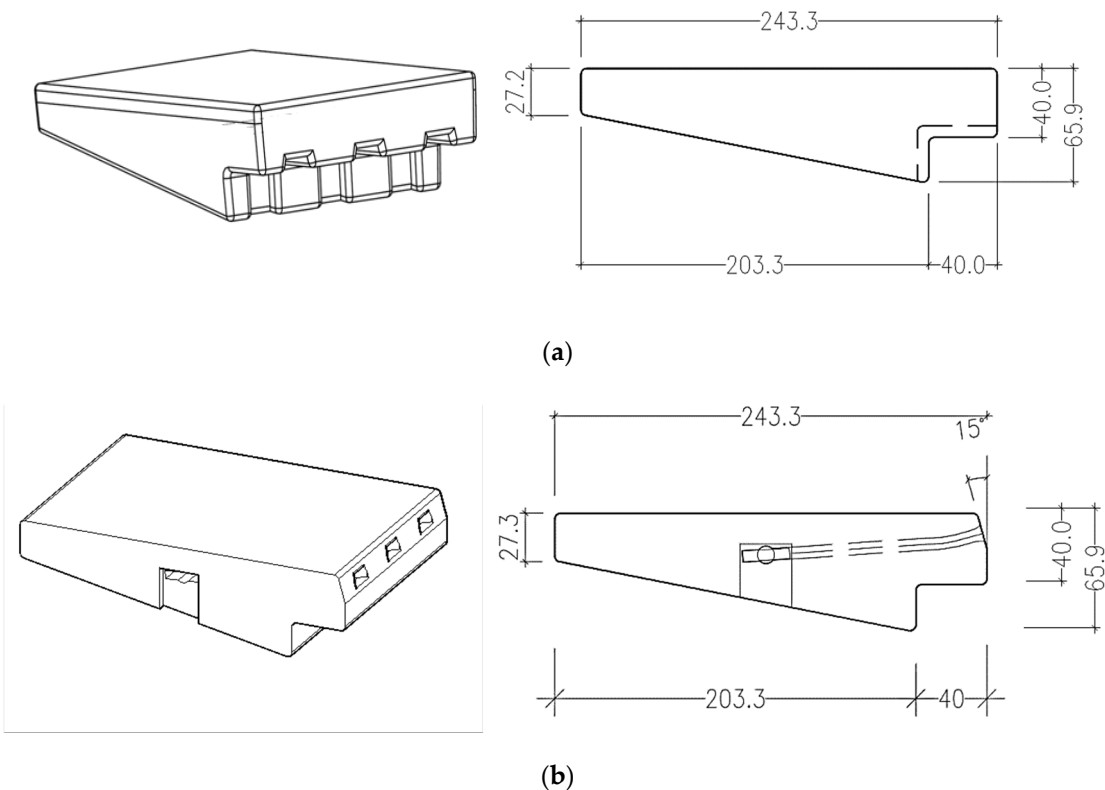

**Figure 7.** A 3D view and longitudinal section of the WSBs tested during the experimental research: (**a**) WSB Armorwedge™ and (**b**) WSB ACUÑA (dimensions in mm, the width of both block types: 165 mm) [30].

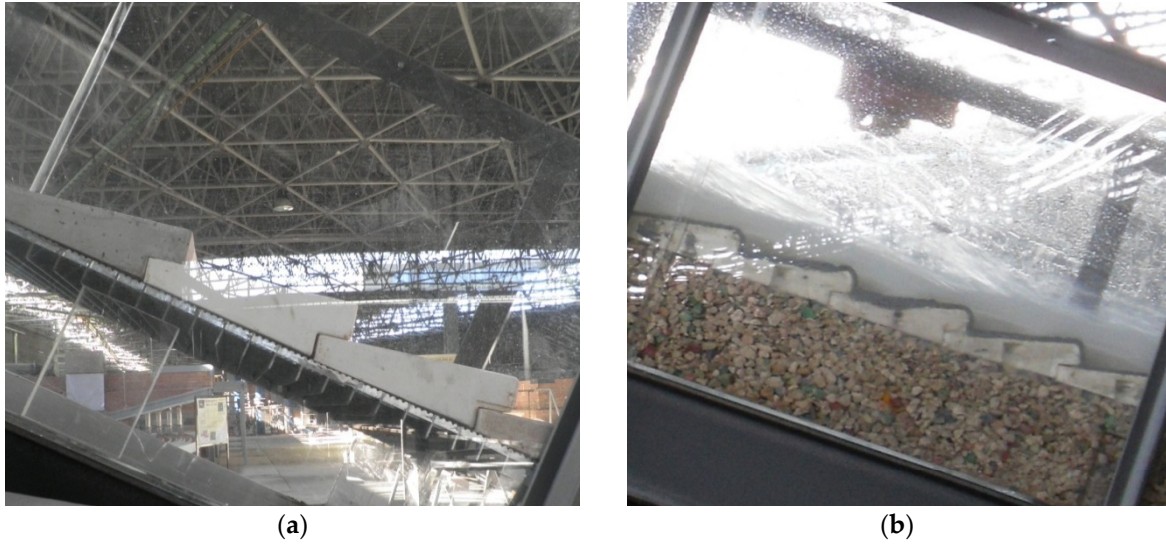

**Figure 8.** WSBs placed over (**a**) a non-slip metallic grid and (**b**) a granular drainage layer.

**Table 4.** A summary of tests carried out (chronologically ordered): Armorwedge[TM] block (*w*1), ACUÑA block (*w*2), Free drainage conditions (*d*1) and Granular layer drainage conditions (*d*2). Measured variables: *P*, hydrodynamic pressures and *L*, leakage flow. Joints and air vents: longitudinal joints (*LJ*), transversal joints (*TJ*) and air vents (*AV*).

| WSB | Drainage Conditions | $q$ (m² s⁻¹) | Measured Variables | Joints and Air Vents Conditions | Number of Tests Performed |
|---|---|---|---|---|---|
| *w*1 | *d*1 | 0.04–0.24 | *P* on rows 5 and 25/*L* | Without sealing | 3 |
| | | | *P* on rows 10 and 30/*L* | Without sealing | 2 |
| | | | *P* on rows 15 and 35/*L* | Without sealing | 2 |
| | | | *P* on rows 15 and 43/*L* | Without sealing | 1 |
| | | | *L* | Without sealing | 1 |
| | | | *L* | *LJ* sealing | 1 |
| | | | *L* | *LJ* and *TJ* sealing | 1 |
| *w*2 | *d*1 | 0.04–0.24 | *L* | Without sealing | 1 |
| | | | | *LJ* sealing | 1 |
| | | | | *LJ* and *TJ* sealing | 1 |
| | *d*2 | 0.04–0.20 | *P* on rows 10 and 25/*L* | Without sealing | 2 |
| | | | | *LJ* sealing | 2 |
| | | | | *LJ* and *TJ* sealing | 2 |
| *w*1 | *d*2 | 0.04–0.20 | *P* on rows 10 and 25/*L* | Without sealing | 1 |
| | | | | *LJ* sealing | 1 |
| | | | | *LJ* and *TJ* sealing | 1 |
| *w*2 | *d*2 | 0.04–0.20 | *L* | Without sealing | 1 |
| | | | | *LJ* and *TJ* sealing of rows 1 to 8. | 1 |
| | | | | *LJ*/*TJ*/*AV* sealing of rows 1 to 8. | 1 |
| | | | | *LJ* and *TJ* sealing of rows 1 to 16. | 1 |
| | | | | *LJ*/*TJ*/*AV* sealing of rows 1 to 16. | 1 |
| | | | | *LJ* and *TJ* sealing of rows 1 to 24. | 1 |
| | | | | *LJ*/*TJ*/*AV* sealing of rows 1 to 24. | 1 |
| | | | | *LJ* and *TJ* sealing of rows 1 to 32. | 1 |
| | | | | *LJ*/*TJ*/*AV* sealing of rows 1 to 32. | 1 |

Then, a set of laboratory tests was performed in order to compare the behavior of ACUÑA and Armorwedge[TM] blocks in two different drainage conditions. First, the free drainage condition (*d*1) was maintained. Thus, the blocks were placed over a fixed metallic grid located 0.2 m over the channel bottom so that there was a free space between the blocks and the base of the channel with the purpose of simulating the conditions of a high permeability underlay (for example, a clean, highly permeable rockfill; Figure 8a). These tests were considered representative of an underlay condition where uplift pressures are not expected. In the second stage, ACUÑA and Armorwedge[TM] blocks were placed over a layer of 0.20 m thick, homogeneous gravel ($D_{50}$ of 12.6 mm, $D_{10}$ of 8.4 mm, $C_u$ of 1.54; Figure 9) layer (*d*2). This granular layer was extended on the impervious bottom of the channel (Figure 8b).

The second set of tests aimed to simulate the hydraulic performance of the blocks over impervious soil, such as clay or sandy clays, with an intermediate permeable bedding layer of gravels.

Conceptually, the presence or absence of saturation of the drainage layer is of special importance in the operation of WSBs. For this reason, additional tests were performed to determine the origin of the drainage flow. The first four rows of WSBs were sealed with the aim of reducing leaks and simulating the first section of a real spillway chute with WSBs and an overlapping slab in the upper area (Figure 1b). The drainage flow was firstly measured with the free drainage condition, *d*1, and then the blocks were placed over a granular drainage layer, *d*2, in order to determine the pattern of the seepage through the blocks towards the drainage layer along the chute with both drainage conditions. Next, the origin of the leakage was investigated with the aim of discriminating between the leaks through the contact joints, which were longitudinal joints and transverse joints in the area

of block overlap, as well as through the vents (Figure 10). These types of tests were carried out both in the Armorwedge^TM and the ACUÑA block. It should be remembered that the blocks were placed in the channel without any type of waterproofing between them.

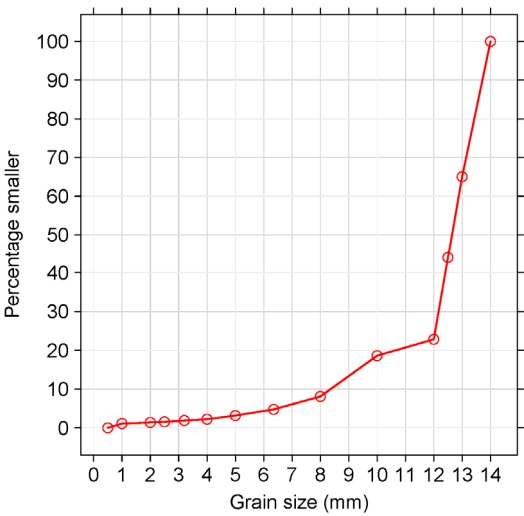

**Figure 9.** The grain size of the drainage layer.

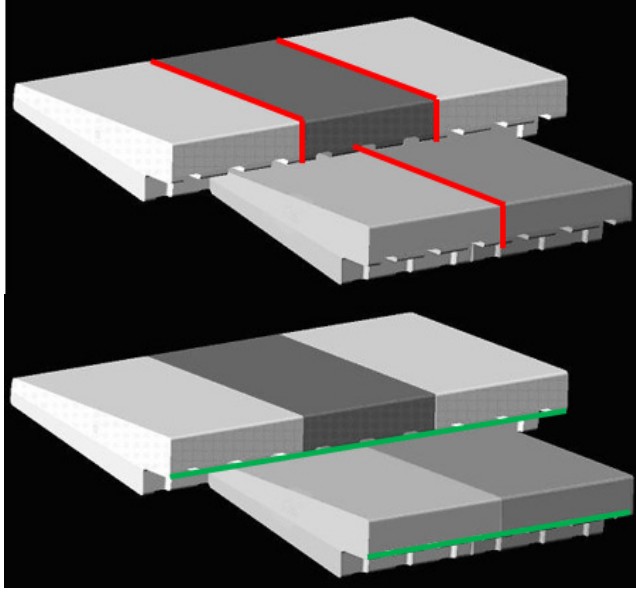

**Figure 10.** Longitudinal joints (red) and transverse joints (green) between blocks. Adapted from [14].

Finally, a new test program was carried out with the ACUÑA block to find out how the leakage flow was distributed among the spillway sectors along the chute. The spillway was completely waterproofed sequentially in the upstream–downstream direction in each of the four sectors of eight rows. The location of the triangular sharp-crested weir allowed the measuring of the drainage discharge up to row 32. A test was also performed with no sealing. The purpose of these tests was to verify whether, as is evident, the greatest leaks were produced by the upper zone of the flume and to obtain the data regarding the percentage. Table 4 presents a summary of the tests carried out for the research described in this article.

Additional photographs and videos of the tests have been included in supplementary materials in the following folders: '03_*w*1_*d*1_2017', '04_*w*2_*d*1_2017', '05_*w*1_*d*2_2018', '06_*w*2_*d*2_2018', '07_*w*2_*d*2_2019_no_sealing' and '08_*w*2_*d*2_2019_sealing_8rows'.

## 3. Results and Discussion

The results obtained in the research regarding hydrodynamic pressures on the different blocks tested ($w$1, Armorwedge™ and $w$2, ACUÑA) and with different drainage situations ($d$1: free drainage conditions and $d$2: blocks placed over a layer of 0.20 m thick homogeneous gravel), as well as the seepage from the flume to the different drainage layers ($d$1 and $d$2), are summarized in the following sections.

### 3.1. Hydrodynamic Pressures

3.1.1. Pressures on the Block Tread

Armorwedge™ Block ($w$1) Tests with Free Draining Conditions ($d$1)

Caballero et al. [28] found good agreement when comparing the registered records of the average pressures with those observed by Bramley et al. [8], Slovensky [11] and Relvas and Pinheiro [19] for a uniform flow regime.

Figure 11 shows additional results for the Armorwedge™ block regarding the average pressure on the sensorized block, which was located in different rows of the stepped chute. The pressure head ($p\,\gamma^{-1}$) refers to the riser height of the block ($h_s$) to obtain a dimensionless parameter ($p\,\gamma^{-1}\,h_s^{-1}$).

In agreement with previous research ([8,11,19]), two pressure zones with positive and negative pressure were found in the block tread. The boundary was located between 30% and 40% of the tread length. The maximum positive pressure heads were registered between 52% and 82% of the tread length. Slovensky registered these as between 52% and 67% of the tread length. However, the magnitude of the standard deviation must be considered when comparing these results. For the skimming uniform flow, the maximum ranges were between 67% and 82% of the tread length. We systematically observed for the skimming flow an increase in the mean maximum pressures on the block tread up to rows 25–30 and a decrease from these rows onwards (Figure 11). In the upper part of the channel, before reaching the uniform flow, the velocity increased, which might explain the increase in the impact pressures in the downstream direction. However, once the uniform flow was reached, the velocity remained constant. The decrease in maximum pressure in this area might be due to the increase in air entrainment [11]. However, this explanation given by Slovensky contradicts the definition of uniform flow. It is possible that uniform flow may not yet be fully achieved, although this fact contradicts existing empirical formulations [3,18,49]. Another possibility would be a measurement error, but as shown in the experimental program (Section 2.3. Table 4), the tests in rows 25, 30 and 35 have been repeated at least 2 times, always obtaining very similar results.

In accordance with [8,11,19], the maximum mean pressures usually increased with an increasing flow rate discharge in all sections in the channel (Figure 11). Some exceptions were observed; for example, in row 25 (Figure 11d), likely due to a measurement error, and in row 10 (Figure 11c), where the average pressures did not increase monotonically, the flow was still accelerated there. Furthermore, as shown in Figure 11, it can be concluded that as the flow moves downstream, there is also a displacement of the mean maximum pressure downstream in the block tread. Thus, for higher flow rates, from 0.16 $m^2\,s^{-1}$ upwards, there is a displacement of the average maximum pressure measured from sensor 5 in row 5 to sensor 3 (Figure 3) in rows 30 and 35. This can be explained by a more distant jet flow launch from the edge of the upper step as a consequence of the acceleration and increase in the speed of the flow. Nevertheless, it should be noted that the exact position of the maximum pressure could not be specified due to the limited number of sensors available.

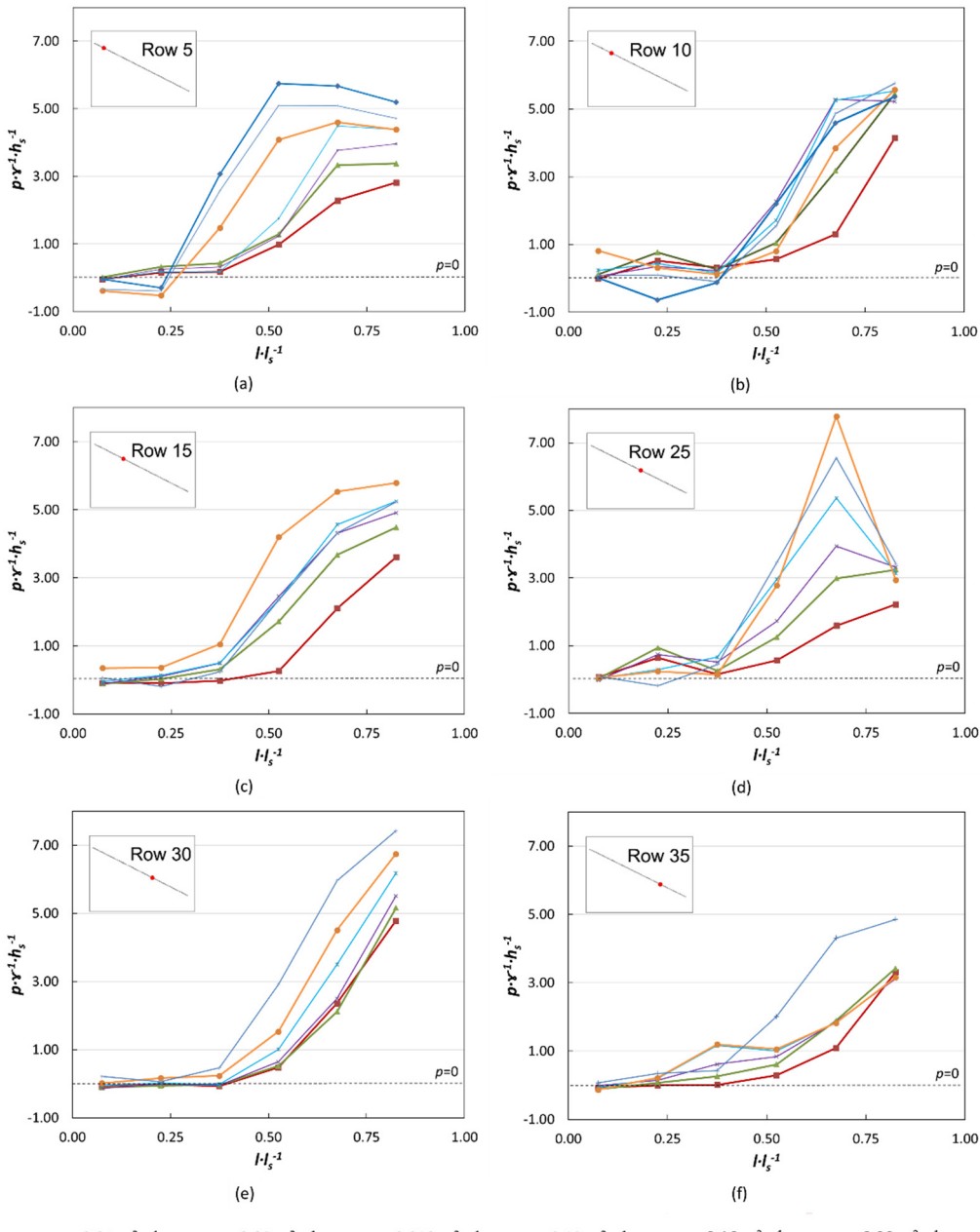

**Figure 11.** Mean pressures on the tread of the Armorwedge™ block (*w*1) in free draining conditions (*d*1). (**a**) Row 5; (**b**) row 10; (**c**) row 15; (**d**) row 25; (**e**) row 30; (**f**) row 35.

A slight displacement of the point between the zone of clearly positive pressures and the zone of negative or close to zero pressures was also observed once the flow accelerated. This aspect also seems quite logical as the impact of the flow on the block tread shifted downstream as the flow accelerated.

Mean pressures and standard deviation on the tread of Amorwedge™ block at different rows of the chute for skimming flow rates have been included in supplementary materials (Figure S5 of Supplementary Materials).

Effect of Drainage Layer (*d*2) on Armorwedge™ Block (*w*1)

The pressures on the block tread are usually similar for free drainage (*d*1) and drainage layer (*d*2) conditions; this was observed on the first four sensors available on the block tread (sensors 8–5 in Figure 3) in row 25 (Figure 12) for all the tested discharge flow rates.

However, the last sensor (sensors 3 in Figure 3) showed greater pressures with up to a 300% increase for the *d*2 conditions (Figure 12b–d). In row 10 (Figure 13), there was good agreement between the values obtained for the *d*1 and *d*2 scenarios in the case of the highest discharge flows on all sensors, except sensor number 3, the last one (Figure 3).

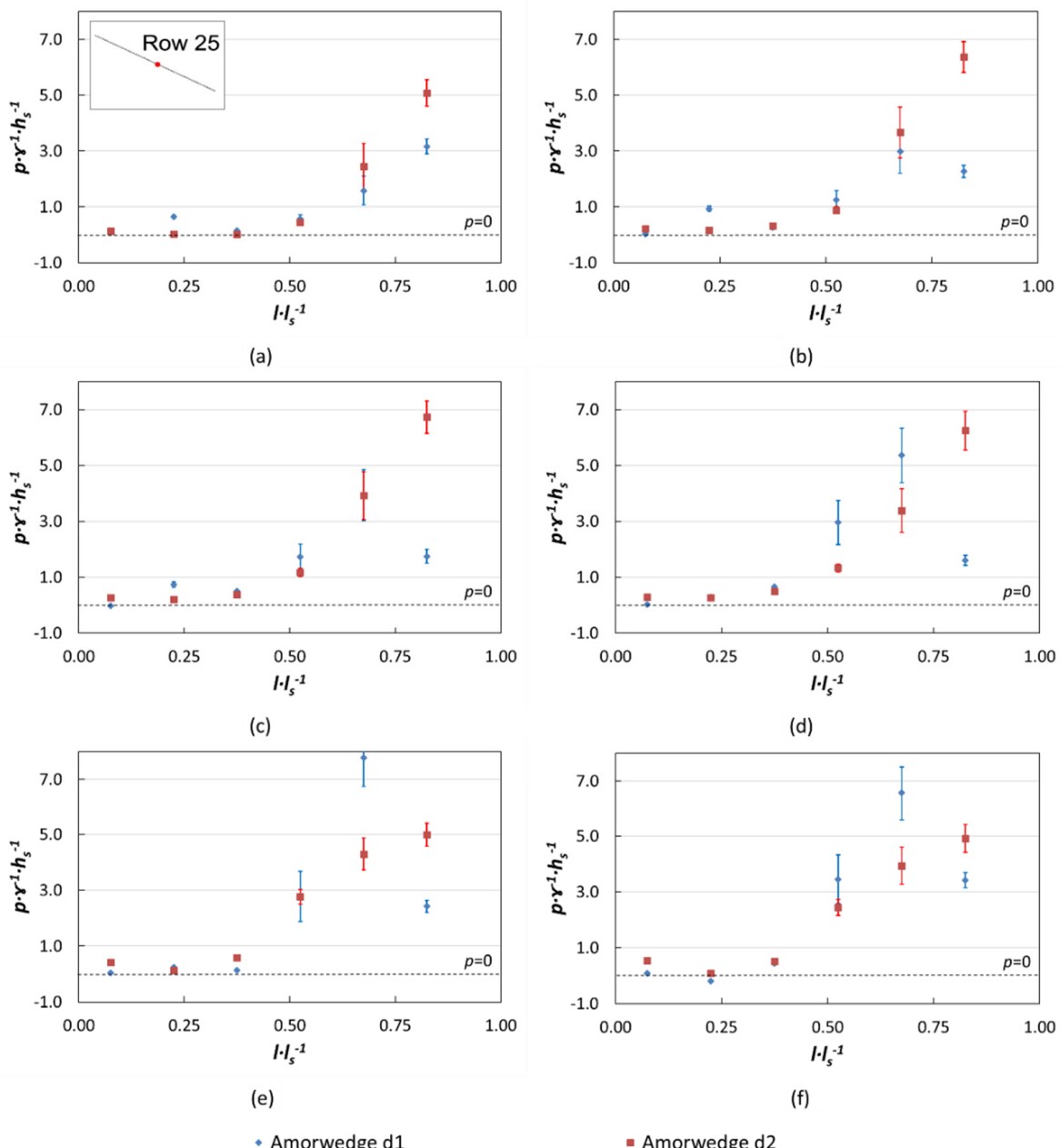

**Figure 12.** Average and standard deviation of the hydrodynamic pressures registered on the tread of Armorwedge$^{TM}$ block (*w*1) in row 25 of the chute in different drainage conditions: free drainage, *d*1, and gravel drainage layer, *d*2, with unit flow: (**a**) 0.04 m$^2$ s$^{-1}$, (**b**) 0.08 m$^2$ s$^{-1}$, (**c**) 0.10 m$^2$ s$^{-1}$, (**d**) 0.12 m$^2$ s$^{-1}$, (**e**) 0.16 m$^2$ s$^{-1}$ and (**f**) 0.20 m$^2$ s$^{-1}$.

The next step was to compare the hydrodynamic pressures on the block tread for the Armorwedge$^{TM}$ and ACUÑA blocks with free drainage and with a granular drainage layer (Figures 14 and 15).

Comparison of the Armorwedge[TM] (*w*1) and ACUÑA (*w*2) Blocks

The comparison was performed with both types of blocks on a drainage layer, which is the usual layout. As a general rule, similar pressures on the tread were observed for the Armorwedge[TM] block (*w*1) and the ACUÑA block (*w*2) in rows 10 and 25 for all flow rates (Figures 14 and 15). Nonetheless, the differences were very modest and barely noticeable; in some sensors, the pressure was slightly higher in the ACUÑA block, in others, it was higher in the Armorwedge[TM] block, and there were also cases where they could be considered almost coincident.

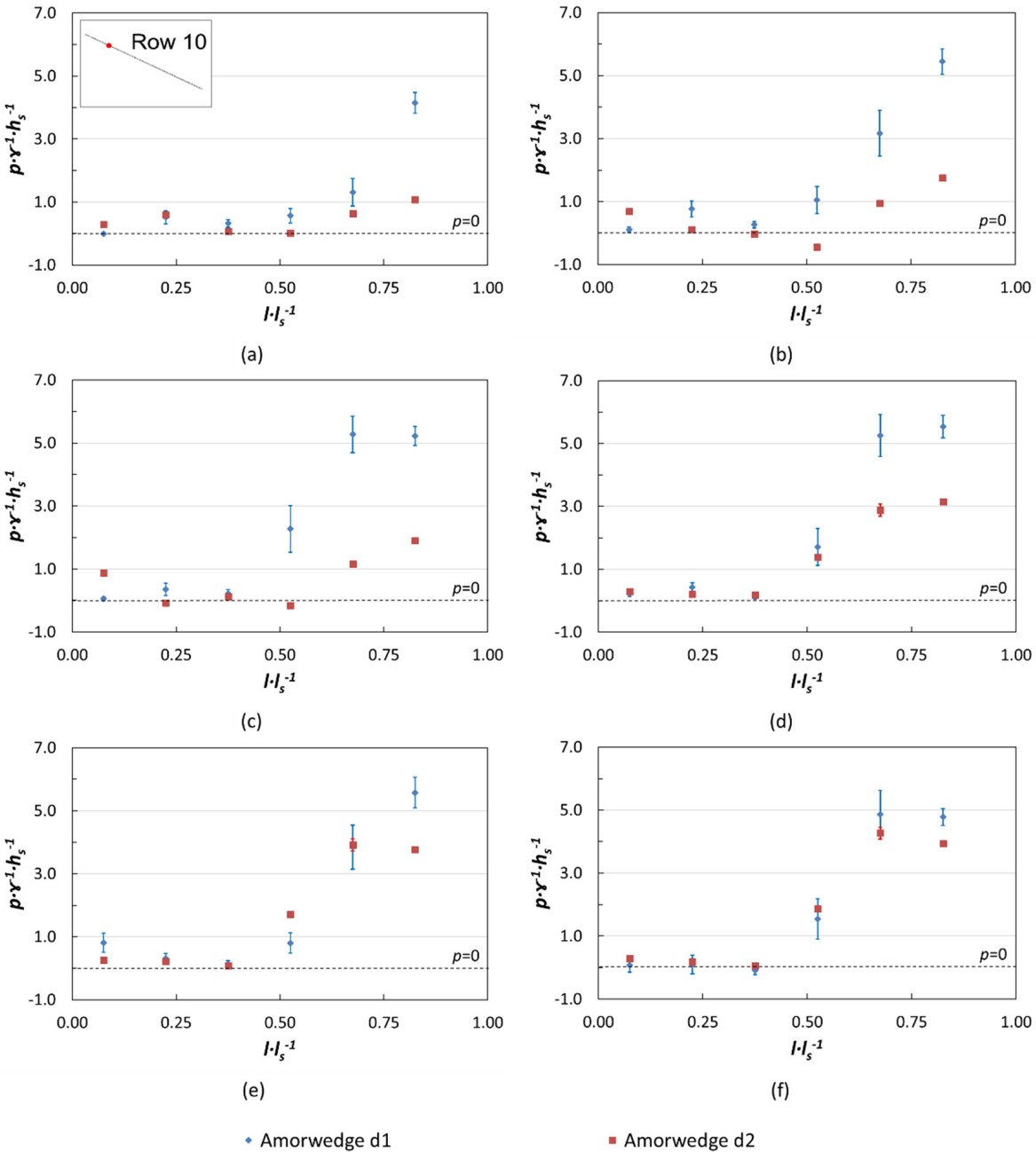

**Figure 13.** Average and standard deviation of the hydrodynamic pressures registered on the tread of Amorwedge[TM] block (*w*1) in row 10 of the chute in different drainage conditions: free drainage, *d1*, and gravel drainage layer, *d2*, with unit flow: (**a**) 0.04 m$^2$ s$^{-1}$, (**b**) 0.08 m$^2$ s$^{-1}$, (**c**) 0.10 m$^2$ s$^{-1}$, (**d**) 0.12 m$^2$ s$^{-1}$, (**e**) 0.16 m$^2$ s$^{-1}$ and (**f**) 0,20 m$^2$ s$^{-1}$.

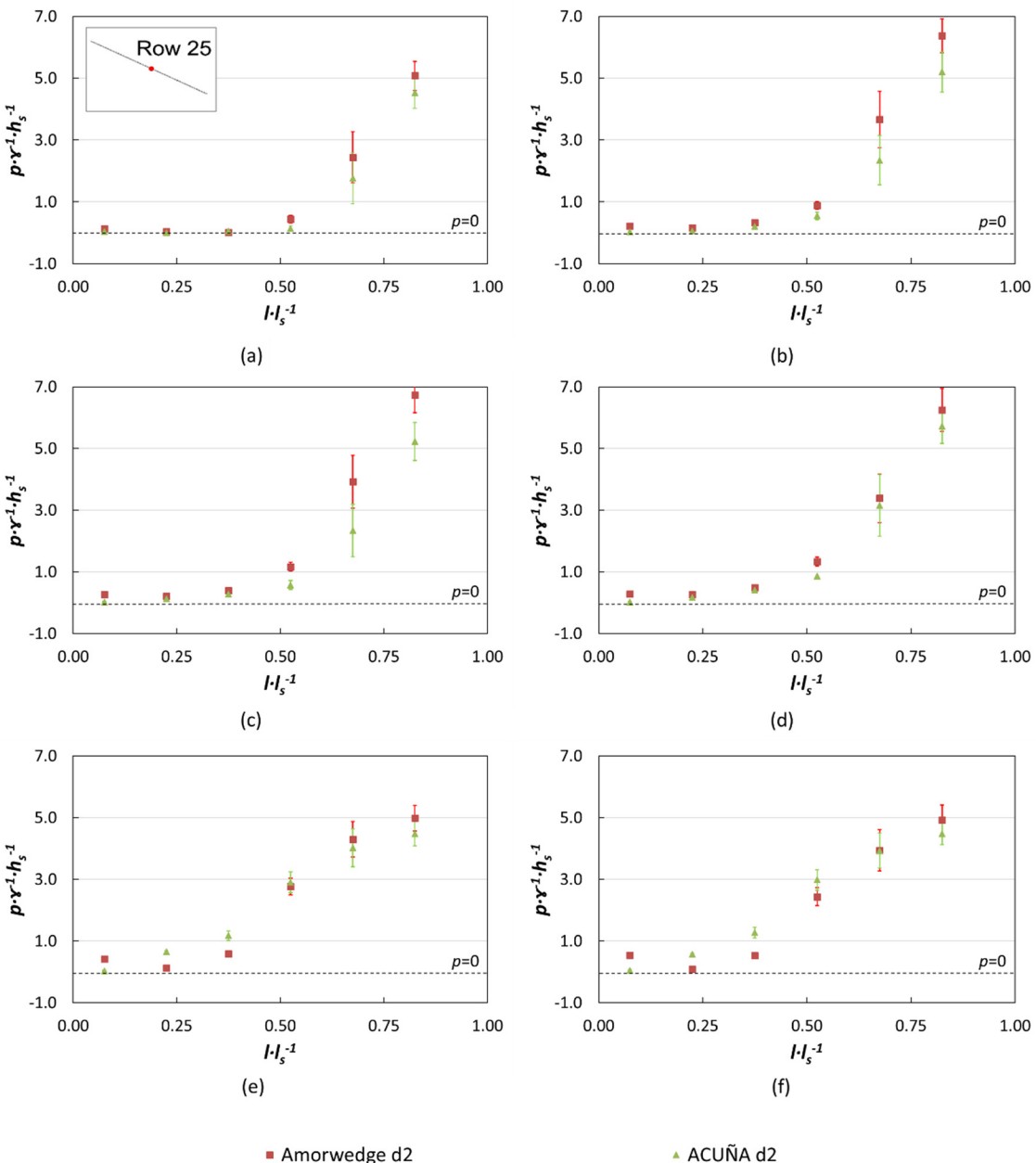

**Figure 14.** Average and standard deviation of the hydrodynamic pressures registered on the tread of Armorwedge[TM] ($w1$) and ACUÑA ($w2$) block in row 25 of the chute for different unit discharges simulating blocks laid over an impervious soil with an intermediate and permeable bedding granular layer ($d2$): (**a**) 0.04 m$^2$ s$^{-1}$, (**b**) 0.08 m$^2$ s$^{-1}$, (**c**) 0.10 m$^2$ s$^{-1}$, (**d**) 0.12 m$^2$ s$^{-1}$, (**e**) 0.16 m$^2$ s$^{-1}$ and (**f**) 0.20 m$^2$ s$^{-1}$.

However, a very slight increase in the maximum pressures for the highest discharged flows (Figure 14) was observed in row 25 for the Armorwedge[TM] block, as well as in row 10 (Figure 15); the point that separates positive and negative, or close to zero, pressures was usually located more upstream in the ACUÑA block. The chamfer of this type of block might move the impact point slightly upward and cause the described effect. This increases the positive pressures on the block tread, which is favorable for block stability.

Effect of Sealing Joints between Blocks

Additional tests were performed for both types of blocks with a drainage layer ($d2$), having previously sealed the joints between the blocks, first the longitudinal ones and then the transverse joints. Although the aim was to discriminate the preferential leakage areas

and quantify the leakage (shown later in this paper), the hydrodynamic pressures were also registered. We observed that there was little variation in the pressures on the block tread compared with the unsealed scenario. Two examples are shown in Figure 16.

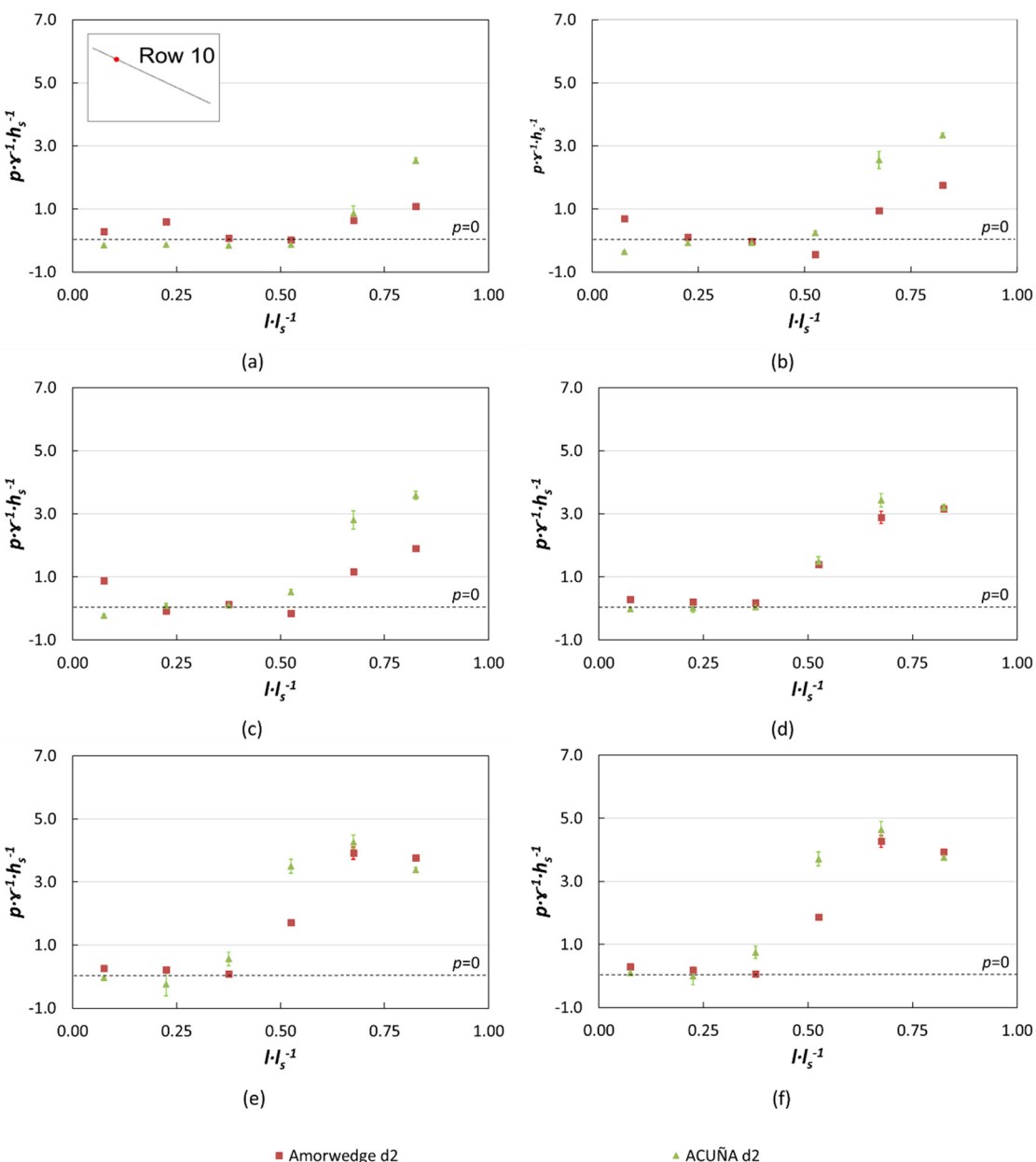

**Figure 15.** Average and standard deviation of the hydrodynamic pressures registered on the tread of the Armorwedge$^{TM}$ (*w*1) and ACUÑA (*w*2) block in row 10 of the chute for different unit discharges simulating the blocks laid over an impervious soil with an intermediate and permeable bedding granular layer (*d*2): (**a**) 0.04 m$^2$ s$^{-1}$, (**b**) 0.08 m$^2$ s$^{-1}$, (**c**) 0.10 m$^2$ s$^{-1}$, (**d**) 0.12 m$^2$ s$^{-1}$, (**e**) 0.16 m$^2$ s$^{-1}$ and (**f**) 0.20 m$^2$ s$^{-1}$.

### 3.1.2. Pressures on the Block Riser

Pressure distribution along the block riser was measured. Several authors ([8,11]) measured the pressure on the block riser at a certain location, but there was a lack of data regarding the pressure distribution along the riser. Thus, the pressure was measured at three points along the riser. As expected, pressures were negative or close to zero at the top of the riser, where there was higher suction (Figure 17).

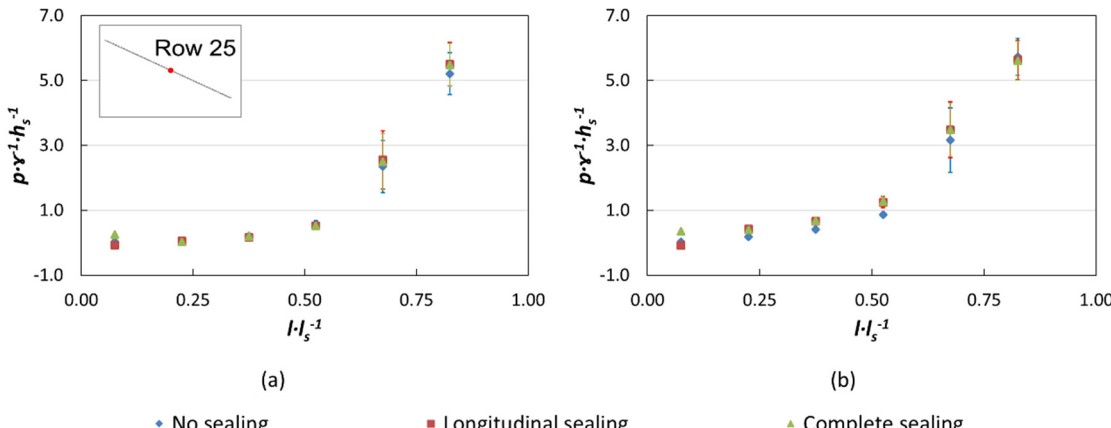

**Figure 16.** Pressure on the block tread of the ACUÑA block ($w2$) in row 25 in scenario $d2$: without joint sealing, with longitudinal sealing and with total sealing. (**a**) 0.08 m$^2$ s$^{-1}$ and (**b**) 0.12 m$^2$ s$^{-1}$.

Similar behavior was observed in the riser of row 25 for both drainage scenarios (Figure 18 and [24], Figure 19), but with different absolute pressure values. Thus, the minimum pressure was systematically located in the upper sensor for all of the tested cases with skimming flow, which was from 0.10 m$^2$ s$^{-1}$ upwards (Figures 18 and 19). Finally, it was also observed that in row 10, the pressure pattern and values were highly coincident for the tests with free drainage ($d1$) and with a drainage layer ($d2$). Figures showing the pressures obtained in the riser are included in supplementary materials (Figures S6–S9 in Supplementary Materials).

The results of the experimental research agree with the results of previous numerical models. Velocity and pressure fields obtained by numerical modeling showed the development of a slightly oscillating vortex with a horizontal axis near the concave junction of the block tread and riser [27,30]. This vortex generates a zone of negative pressure on the surface of the riser, with minimum values in the upper third [27,30] near the edge of the riser. This flow pattern was also observed in the experimental test. The characteristics of the vortex determine the pressure distributions in the tread and riser of the block as well as the operating conditions of the vents. The vents of the Armorwedge$^{TM}$ block are located at the base of the riser. Positive pressure in that area, although low, might cause a flow circulation towards the drainage layer through the vents. Moving the position of the air vents on the ACUÑA block to the upper part of the riser was proposed after observing the described flow pattern and the values of the pressures on the base of the block, shown in the next section.

### 3.1.3. Pressures on the Base of the Block

Three pressure gauges were installed at the base of the block (Figure 3). Two of them registered similar values in agreement with [8], while the third seemed to have not been measured correctly. Therefore, we decided to evaluate the pressure on the base of the block, assuming a uniform distribution with an average value as registered by one of the two sensors that measured similar pressures.

Armorwedge$^{TM}$ Block ($w1$) Tests in Free Draining Scenario ($d1$)

The free drainage conditions allow the complete evacuation of leakage, avoiding the saturation of the drainage layer and facilitating the transmission of suction to the base of the block, which was measured here for the first time. Hydrodynamic pressures at the base of the Armorwedge$^{TM}$ block decreased as the unit discharge of skimming flow over the block increased (Figure 20). At the end of the test, the flow discharge decreased to zero. As shown in Figure 20, the base of the block was mainly subjected to sub-atmospheric pressures.

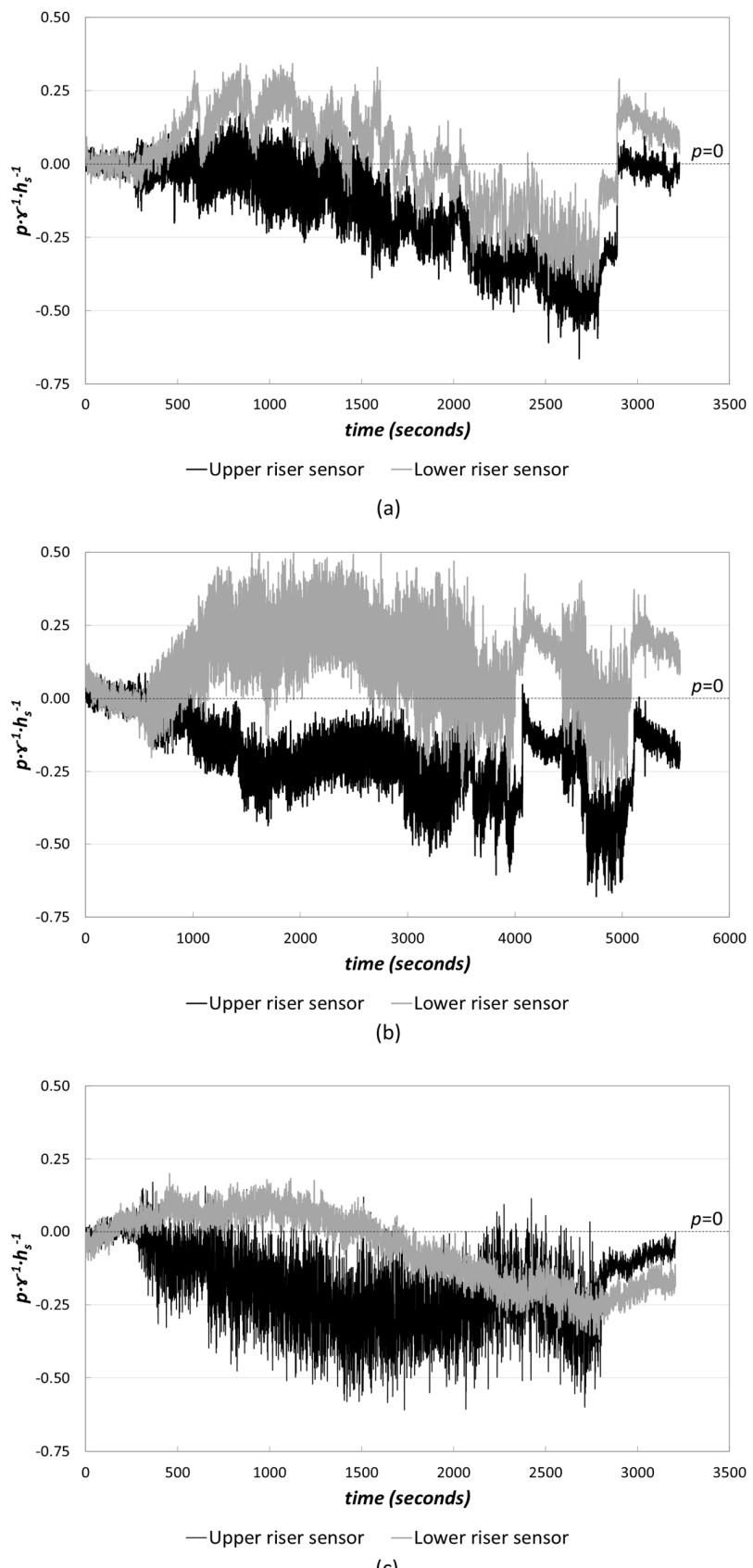

**Figure 17.** Hydrodynamic pressures on the Armorwedge$^{TM}$ block (w1) riser in rows: (**a**) 15; (**b**) 25; (**c**) 35 for unit flow rates (q). They range from 0.04 m$^2$ s$^{-1}$ to 0.24 m$^2$ s$^{-1}$ and a decrease to zero discharge.

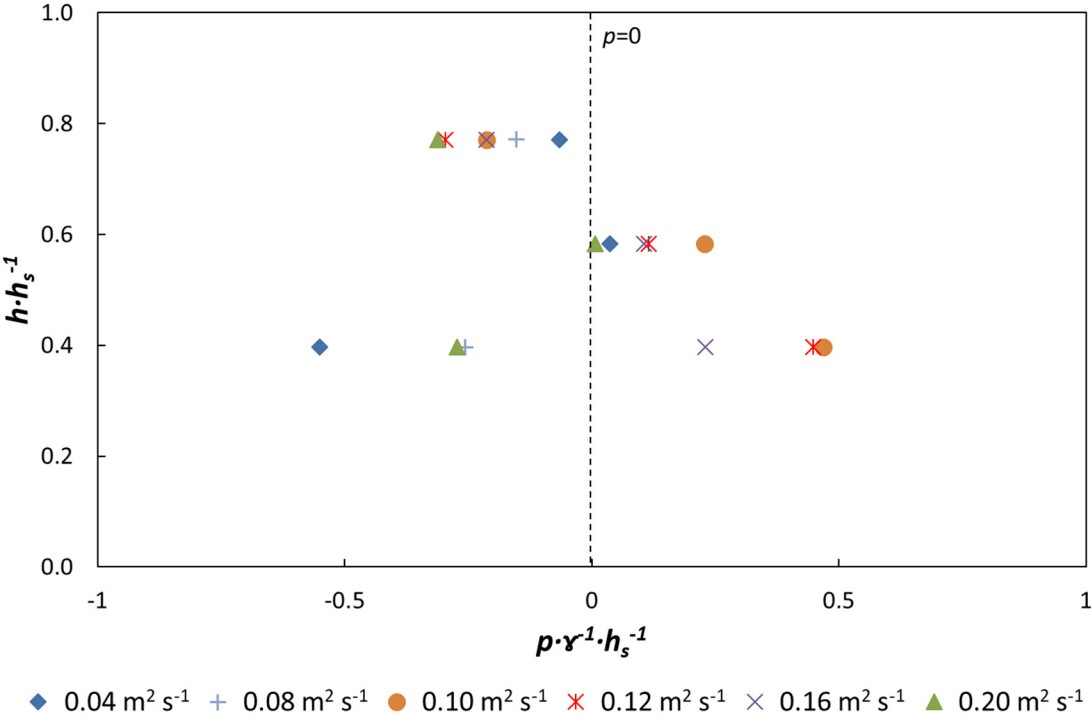

**Figure 18.** The average of the hydrodynamic pressures registered on the block riser of the Armorwedge[TM] block (*w*1) in row 25 of the chute for different unit discharges simulating free drainage conditions (*d*1).

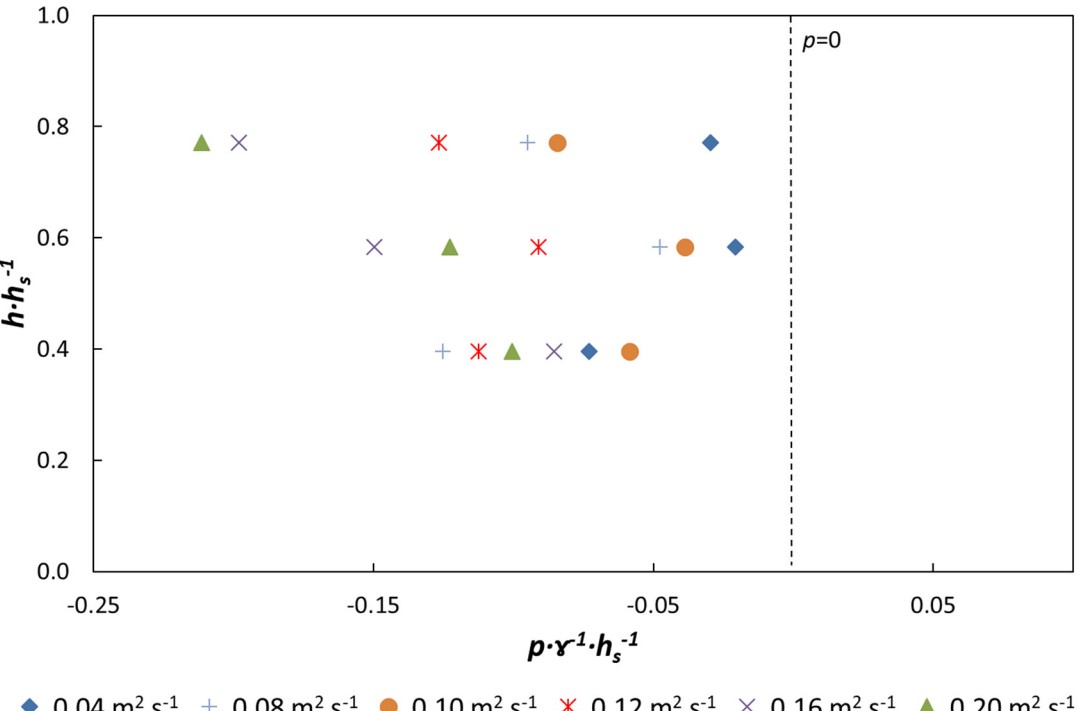

**Figure 19.** The average of the hydrodynamic pressures registered on the block riser of the Armorwedge[TM] block (*w*1) in row 25 of the chute for different unit discharges simulating the blocks laid over an impervious soil with an intermediate and permeable bedding granular layer (*d*2).

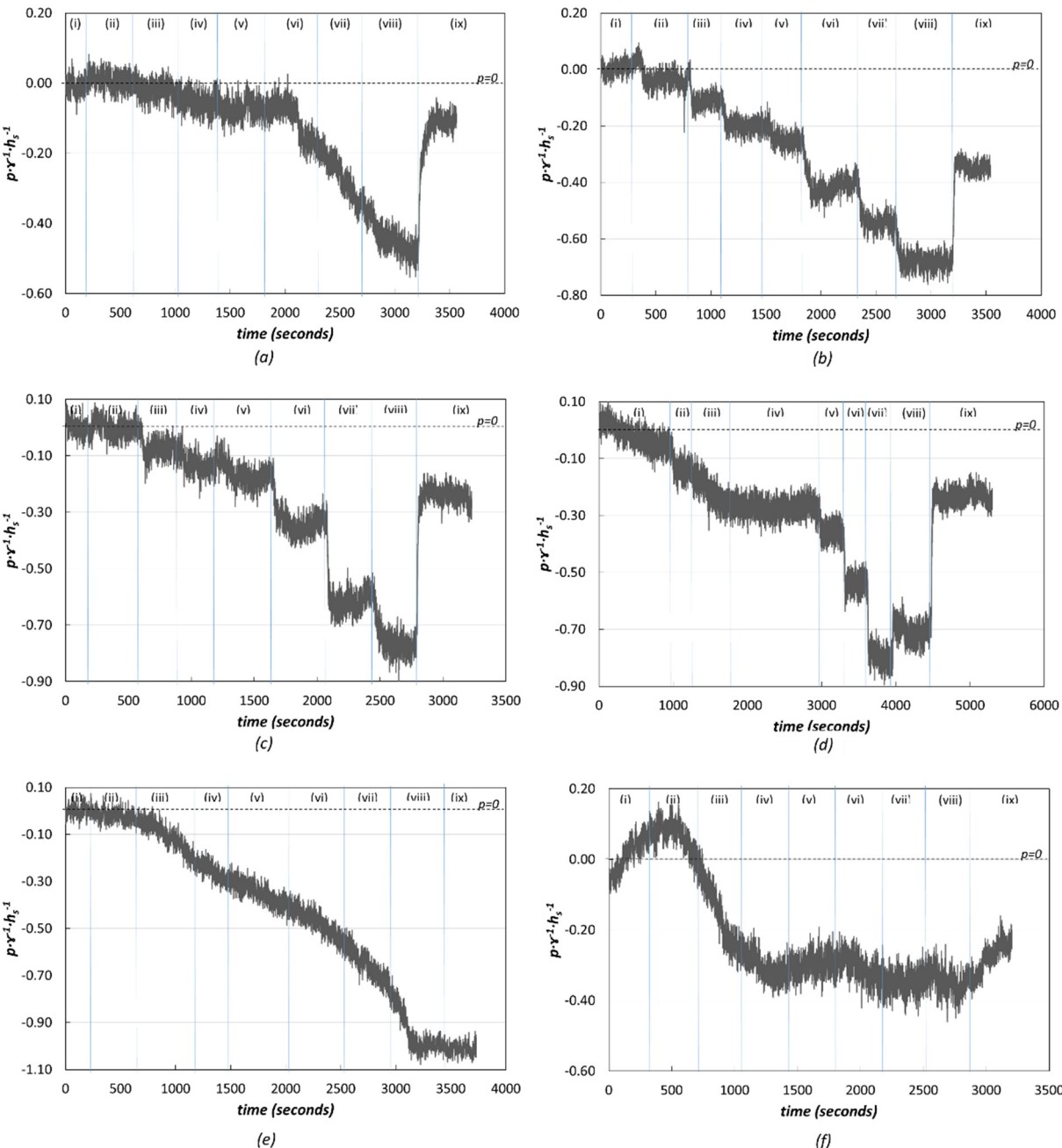

**Figure 20.** Hydrodynamic pressures on the base of the Amorwedge$^{TM}$ block ($w1$) in rows 5 (**a**), 10 (**b**), 15 (**c**), 25 (**d**), 30 (**e**) and 35 (**f**) for unit flow rates ($q$): (i) 0 m$^2$ s$^{-1}$, (ii) 0.04 m$^2$ s$^{-1}$, (iii) 0.08 m$^2$ s$^{-1}$, (iv) 0.10 m$^2$ s$^{-1}$, (v) 0.12 m$^2$ s$^{-1}$, (vi) 0.16 m$^2$ s$^{-1}$, (vii) 0.20 m$^2$ s$^{-1}$, (viii) 0.24 m$^2$ s$^{-1}$ and (ix) 0 m$^2$ s$^{-1}$.

It was observed in the test trials that the time required to reach the stationary state at a given flow rate amounted to very few seconds. However, some measurements were observed where this did not occur, such as in row 30 (Figure 20e) and with some of the flow rates in row 5 (Figure 20a) and 35 (Figure 20f). In general, it was also observed that the return to zero pressure did not occur when the flow discharge decreased to zero, an aspect that may be due to a hysteresis phenomenon. In most of the situations analyzed (Figure 20a–d), this phenomenon was modest; however, it was very prominent in rows 30 (Figure 20e) and 35 (Figure 20f). Nonetheless, it should be noted that the tests were repeated at least twice for each row, obtaining very similar results.

Comparison of the Armorwedge$^{TM}$ (w1) and ACUÑA (w2) Blocks

Both types of blocks were placed on a drainage layer. The results for the ACUÑA and Armorwedge$^{TM}$ block are shown in Figure 21. Positive uplift was registered with a lower flow rate discharge for the Armorwedge$^{TM}$ block (0.12 m$^2$ s$^{-1}$) than for the ACUÑA block (0.16 m$^2$ s$^{-1}$).

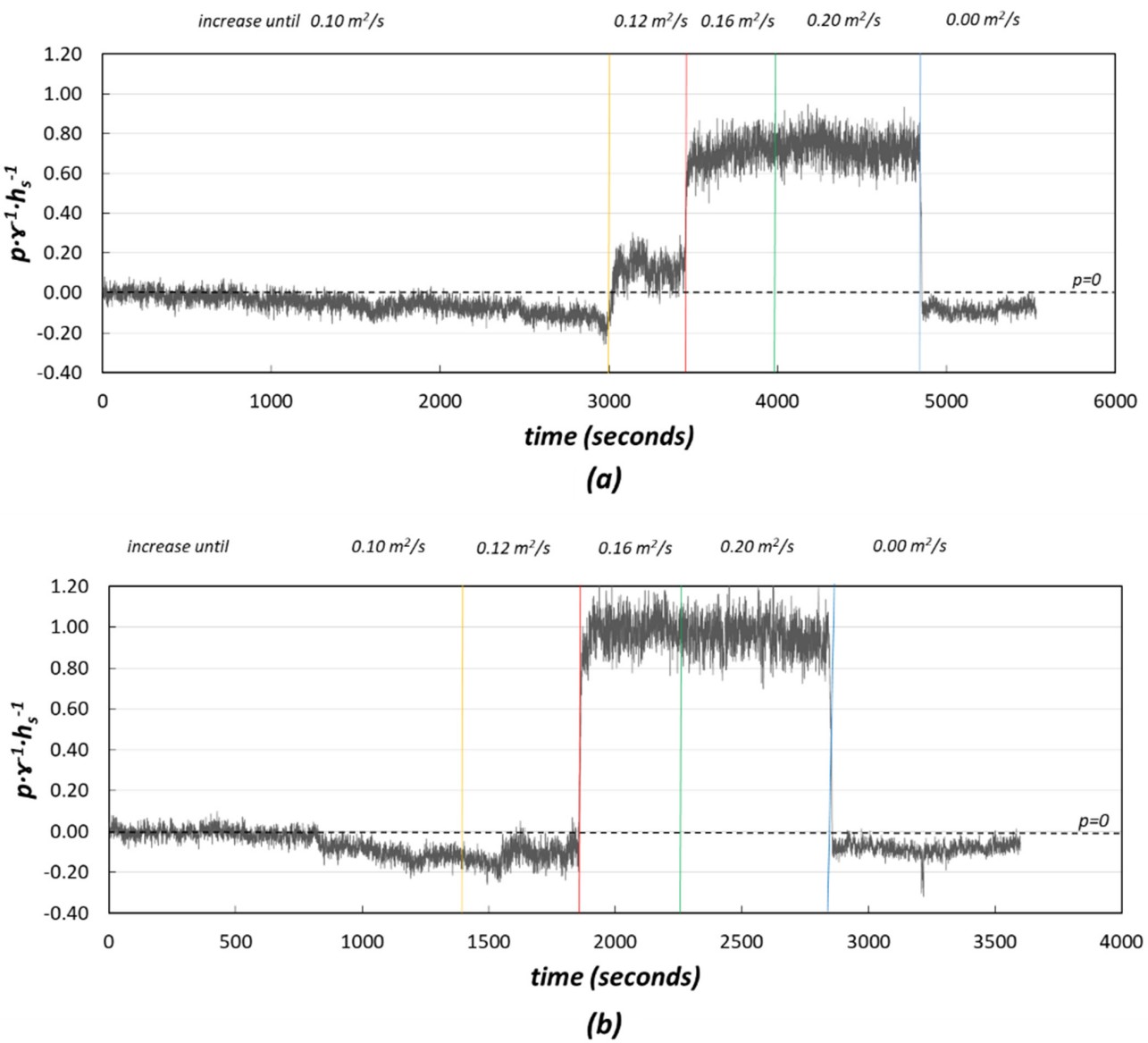

**Figure 21.** Hydrodynamic pressures on the base of the Armorwedge$^{TM}$ block (w1) (**a**) and ACUÑA block (w2) (**b**) in row 25 for unit flow rates (q). Pressures range from 0.04 to 0.24 m$^2$ s$^{-1}$ and a decrease to zero discharge.

The drainage layer was not saturated for the flow rate of 0.12 m$^2$ s$^{-1}$, whereas it was for the Armorwedge$^{TM}$ block. This might be due either to a lesser drainage flow from the chute towards the drainage layer or to a greater capacity of leakage reintegration from the drainage layer to the chute as a consequence of increased negative suction pressure through the vents. This effect was significantly greater when the joints between blocks were sealed (Figure 22). However, when the drainage layer became saturated, the uplift on the base of the ACUÑA block was greater than that on the Armorwedge$^{TM}$ block for the same discharge value. It is possible that the air vents of the Armorwedge$^{TM}$ block, being in the lower part of the block riser, facilitated the outlet of the water contained in the drainage layer better than those of the ACUÑA block and, therefore, the relief of uplift pressures.

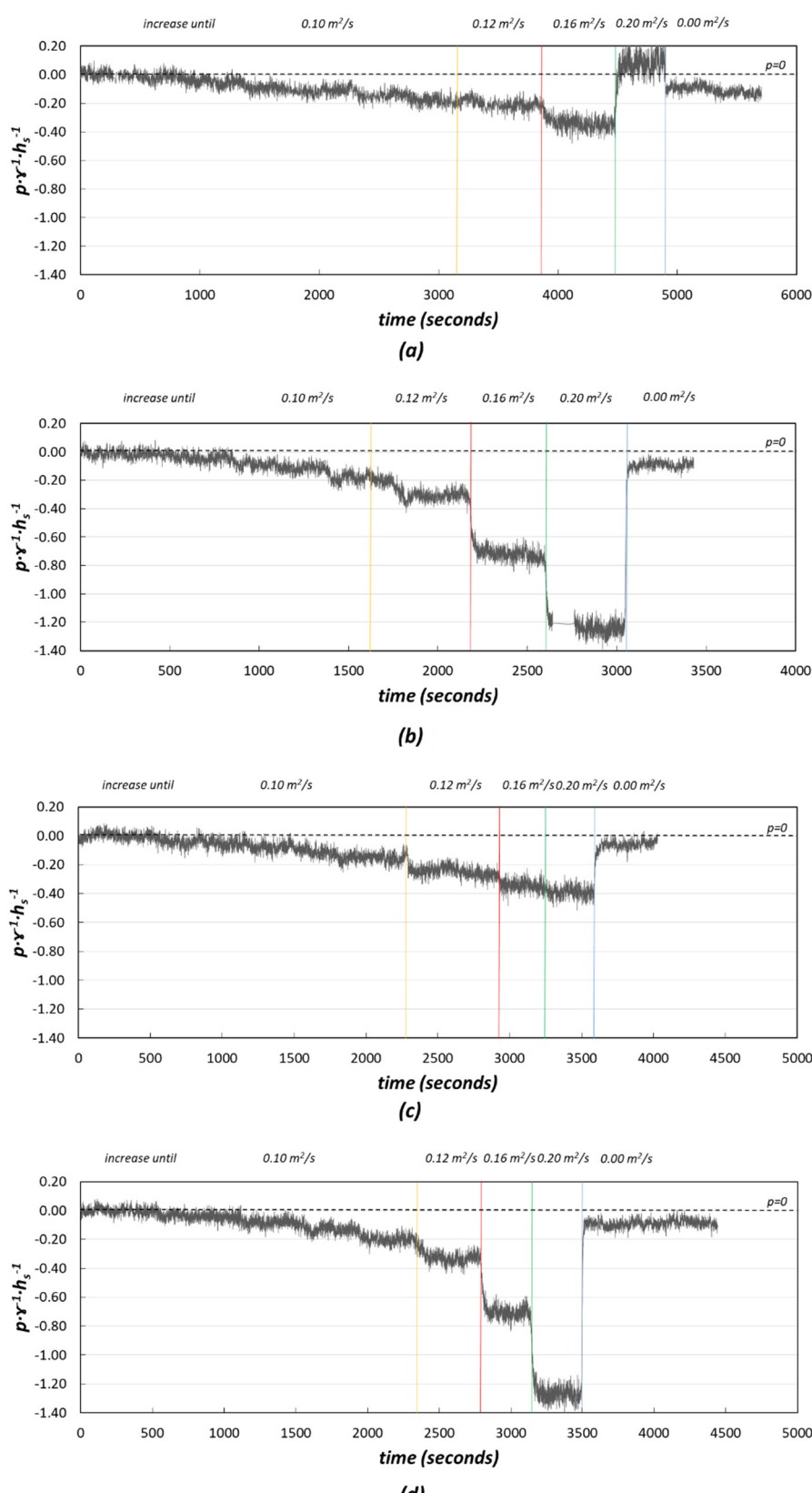

**Figure 22.** Hydrodynamic pressures on the base in row 25 for unit flow rates (*q*) from 0.04 to 0.24 m² s⁻¹ and return to non-discharge. Longitudinal joints sealing: (**a**) Armorwedge™ block (*w*1) and (**b**) ACUÑA block (*w*2). Complete sealing of the joints: (**c**) Armorwedge™ block and (**d**) ACUÑA block.

*3.2. Hydrodynamic Forces on the WSBs*

Hydrodynamic forces play a critical role in the stability of WSBs. They can be classified as the following: forces due to the impact of the main discharge flow on the tread of the block, suction forces developed as a consequence of negative pressures generated in the block tread immediately downstream of each step, uplift on the base due to the saturation of the drainage layer ([8,11,50]), and suction force on the block base if the drainage layer is not saturated. The magnitude of the forces varies with the discharge and block location along the chute. Assuming the same block weight, the difference is negligible; the type of WSB with a more favorable hydrodynamic resultant force will be more stable. The two main hydrodynamic forces acting on the blocks with a saturated drainage layer are the force due to the jet flow impact on the block tread and the uplift force on the base of the block [13]. A pressure field on the block tread was defined based on the pressure measurements (Figures 14 and 15). Uniform distribution with the uplift pressure mean values registered was considered appropriate for the base of the blocks. The difference between total forces on both faces was the resultant hydrodynamic force on the WSB. A comparison of the two types of WSBs, Armorwenge[TM] and ACUÑA, was carried out in three different scenarios considering the sealing of the joints between blocks: no sealing, longitudinal sealing and complete sealing (longitudinal and transversal), as described in Section 3.1.3.

Figure 23 shows the resultant hydrodynamic force per unit width on Armorwedge[TM] and ACUÑA WSBs in rows 10 and 25 as a function of $h_c h_s^{-1}$. The positive values correspond to stabilizing forces. It is relevant to note that the blocks were inherently stable due to the hydrodynamic force, even without considering the contribution of the weight of the block. The weight of the block per unit width was 272 N/m. The hydrodynamic resultant force ranged between 13% and 59% of the weight of the block in cases without sealing of the joints and between 11% and 99% with sealed joints. In row 10, in the upper part of the chute, the stabilizing force systematically increased with discharge flow rate, and the ACUÑA block was more stable than Armorwedge[TM]. This happened independently of the condition of the joint, sealed or unsealed. In row 25, in the lower part of the chute with fully developed velocity and with unsealed joints, the stabilizing force was reduced with a discharge flow rate equal to or greater than 0.12 $m^2$ $s^{-1}$ ($h_c h_s^{-1}$ = 2.84) on the Armorwedge[TM] block. The limit value was slightly higher (0.16 $m^2$ $s^{-1}$ or $h_c h_s^{-1}$ = 3.44) for the ACUÑA block. The cause was undoubtedly the uplift due to the saturation of the drainage layer. The Armorwedge block was more stable than the ACUÑA block with the saturated drainage layer (Figure 23a). However, in row 25 in the two scenarios with sealed joints (Figure 23b,c), the ACUÑA block systematically increased its stability with the flow rate discharge. Additionally, for the flow rates in the skimming flow regime, the ACUÑA block was more stable than the Armorwedge[TM] block.

Although, in all cases the blocks were inherently stable (even more so considering that their weight and the force generated by the interlocking between blocks were not taken into account in the presented calculation), the discussion concerns whether it is realistic for joint sealing to be envisaged in design practice or not. This action is considered viable since, as has been demonstrated, the sealing of the joints (mainly the longitudinal ones and in the upper part of the flume, as presented in Section 3.4) can be very useful for better performance of the technology. Nonetheless, the use of precast blocks that already incorporate waterproofing strips in the contacts between the joints could also be considered.

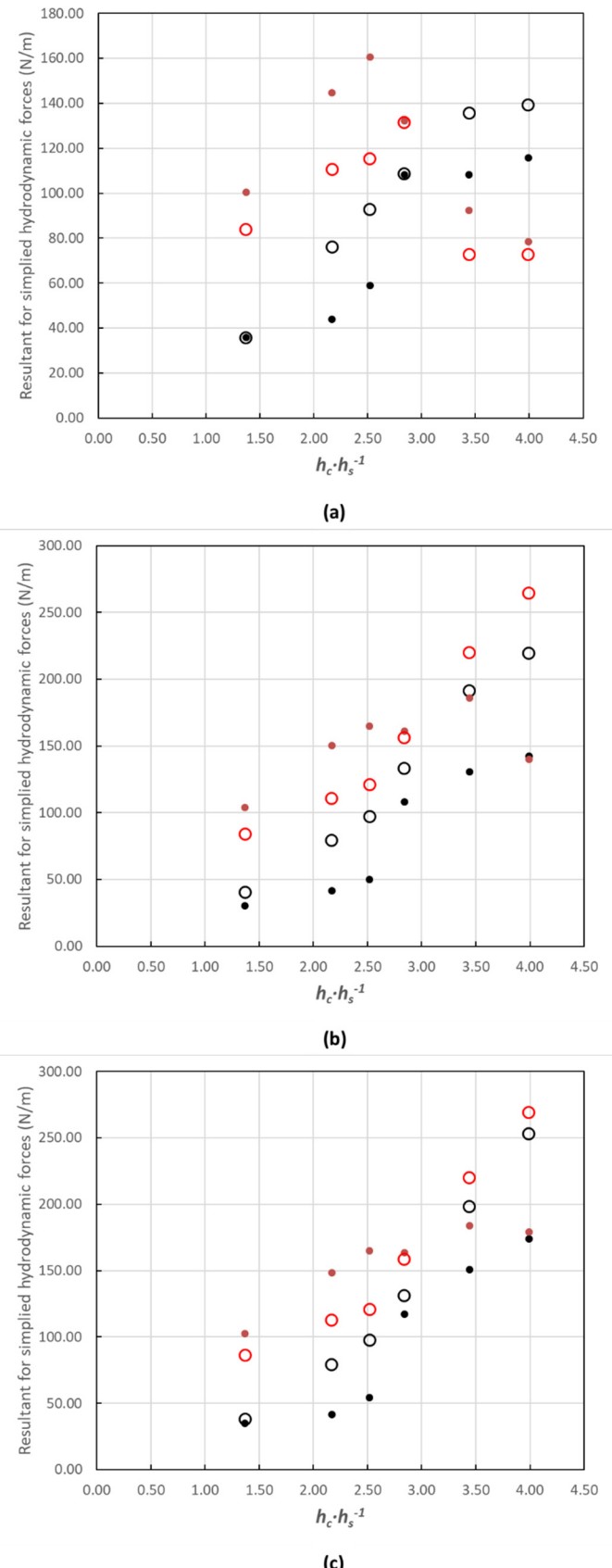

**Figure 23.** Resultant of the hydrodynamic forces on the block tread and base for the Armorwedge[TM] (*w*1) and ACUÑA (*w*2) block for unit flow rates (*q*) from 0.04 ($h_c\,h_s^{-1}$ = 1.37) to 0.24 m$^2$ s$^{-1}$ ($h_c\,h_s^{-1}$ = 3.99): (**a**) joints unsealed; (**b**) longitudinal joints sealed; (**c**) longitudinal and transversal joints sealed.

### 3.3. Drainage Flow

The registered drainage flow rates (Figure 24) were compared with the results of the tests performed by Relvas and Pinheiro [16,18]. Their results show a slight reduction in the drainage flow as the inlet discharge of the tests increases. In our experiments, the drainage flow rate shows a clear difference depending on the drainage configuration: free drainage (*d*1) or with a granular drainage layer (*d*2).

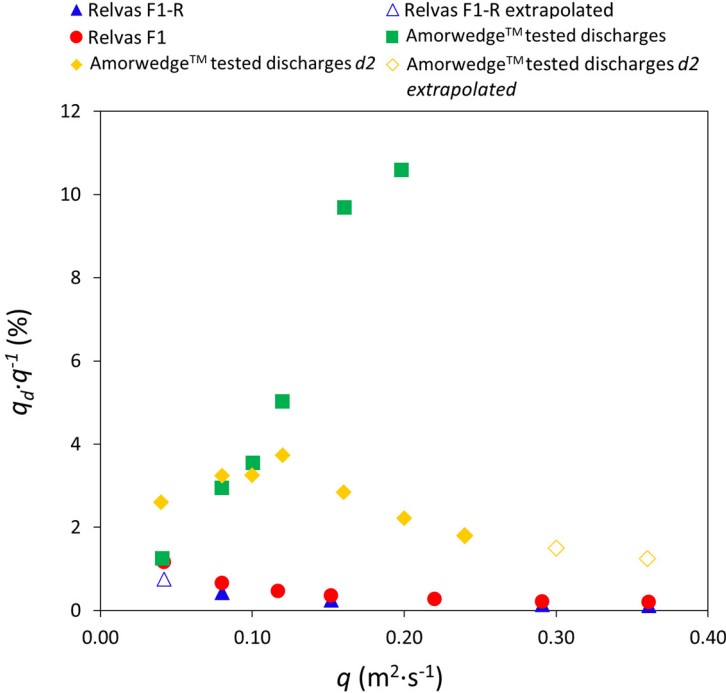

**Figure 24.** The unit drainage flow ($q_d$) expressed as a percentage of the inlet flow rate ($q$).

Configuration *d*1 prevents the saturation of the drainage area and allows the complete evacuation of the leakage flow through the joints. In this situation, unlike the results reported by Relvas and Pinheiro, there is an increase in the percentage of drainage flow as the main discharge flow becomes higher, reaching values up to 10% for the higher flow rates tested (0.20 m² s⁻¹) (Figure 1). However, the tests made with the blocks placed over a granular drainage layer (*d*2) showed a different behavior of the drainage unit flow, with evolution more similar to that described by Relvas and Pinheiro. Nevertheless, the values of the drainage flow rates ($q_d/q_i$) of the configuration *d*2 present a significant difference compared with those observed by Relvas and Pinheiro. This might be due to the different particle sizes of the materials of the drainage layers. Relvas reported a 0.20 m thick drainage layer that integrates two sublayers: a 0.15 m thick upper layer in contact with the concrete blocks and gravel varying between 4/6 ($D_{50}$ of 5 mm, $D_{10}$ of 2.6 mm, *Cu* of 2.3) and10/20 ($D_{50}$ of 14.9 mm, $D_{10}$ of 10.7 mm, *Cu* of 1.5) and a 0.05 m thick bottom layer of sand ($D_{50}$ of 0.8 mm, $D_{10}$ of 0.3 mm, *Cu* of 2.9) [17].

The different drainage flow rates between configurations *d*1 and *d*2 might be due to the degree of saturation underneath the WSBs. Thus, configuration *d*2 can be completely saturated when the maximum seepage capacity of the drainage layer is reached. At that point, once the drainage layer is saturated, the higher the discharge flow, the lower the drainage flow rate. In this situation, when the negative pressure is developed in the vents at the base of the riser of each block, a fraction of the drainage flow is expected to be sucked upward towards the main flow area.

### 3.4. Effect of the Joints among WSBs on the Drainage Flow

The set of tests with and without sealing of the joints allowed us to determine the origin of the drainage flow. A clear predominance of the leakage flow through the longitudinal joints was observed. The longitudinal joints were the origin of 55–80% of the total drainage flow; the higher percentage corresponds to the lower flow rate. The rest of the leakage was produced by both of the horizontal transverse joints between the blocks and the air vents. These results suggest that a reduction in the length of longitudinal joints could significantly reduce the leakage flows towards the drainage layer. The quantity of total drainage flow was quite similar for both types of WSB, Armorwegde[TM] and the proposed ACUÑA. This is logical, taking into account the significant predominance of leakage through the longitudinal joints.

The registered leakage flow rates through the air vents of the ACUÑA block were negligible for flow rates lower than 0.1 $m^2$ $s^{-1}$. Such drainage flow rates increased up to 1–2% of the total leakage for higher inlet flow rates, reaching the maximum value for the inlet discharge of 0.24 $m^2$ $s^{-1}$. Thus, the tests conducted with the ACUÑA block showed a reduction in the total drainage flow rate, compared with the Armorwedge[TM] block, from 12–15% for inlet flow rates between 0.04 and 0.24 $m^2$ $s^{-1}$, respectively. The reduction in the drainage flow rate of the ACUÑA block might explain the relief of the uplift pressures (Figures 21 and 22) compared with those registered for the Armorwedge[TM] WSB.

Additional tests were performed to find out how the leakage flow rate was distributed along the chute for the ACUÑA WSB (Figure 25). Most of the drainage flow originated in the first part of the chute (the first eight rows, see folder "07_*w*2_*d*2_2019_no_sealing" in supplementary materials), so the sealing of the first quarter of the chute may significantly reduce the flow towards the drainage layer (see folder "08_*w*2_*d*2_2019_sealing_8rows" in supplementary materials") for the tested flow rates (up to 0.12 $m^2$ $s^{-1}$). In fact, it was observed that the effect of the sealing has a greater impact on the upper area of the chute. The sealing of the first four rows (Figure 24) led to very similar results to those obtained with the sealing of the first eight rows. The low velocity of the main flow in the upstream area of the chute probably favors the leakage through the joints among the blocks.

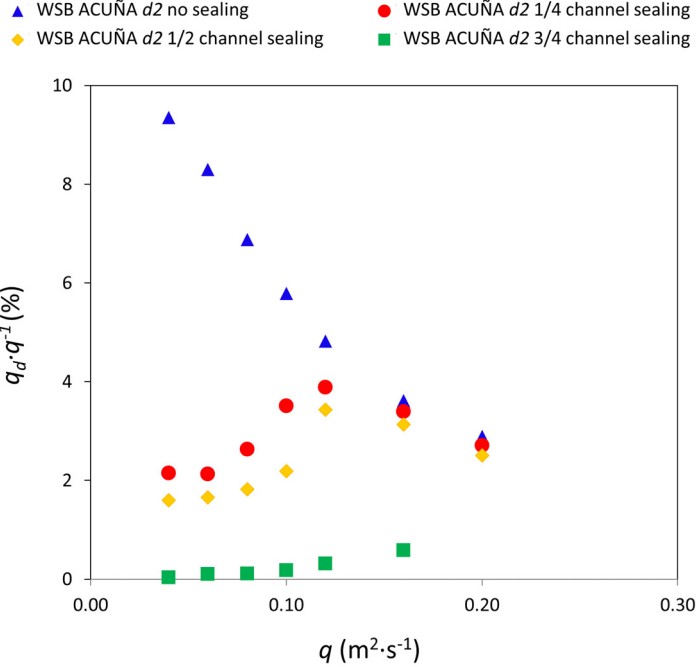

**Figure 25.** The unit flow rate through the drainage layer ($q_d$) expressed as a percentage of the flow rate ($q$) for different $q$ and joint-sealing situations.

## 4. Conclusions

A reference WSB, the Armorwedge™ block, and a new WSB, the ACUÑA block, were tested. The tests were performed in two different drainage conditions: free drainage, evacuating all the leakage flow towards the drainage layer without reaching saturation conditions and drainage through a granular layer with limited hydraulic capacity.

The main conclusions are as follows:

- Hydrodynamic pressures on the blocks tread were similar for the Armorwedge™ and ACUÑA blocks, although a slightly higher pressure was observed on the ACUÑA block for the highest discharge flows in the lower part of the chute. Although a limited effect, this is favorable for the stability of the block.
- Pressure records in the riser of the ACUÑA block were negative or close to zero, with the greatest suction located in the upper third of the riser. This fact was also previously observed by numerical modeling and led to the new WSB design, ACUÑA, with air vents in the upper part of the riser.
- Negative suction pressures were registered at the base of the two types of blocks when the drainage layer was not saturated. This is favorable for the stability of the block. The suction at the base was higher when the longitudinal joints between blocks were sealed. The effect of sealing just the upper part of the chute was remarkable. The leakage towards the drainage layer was significantly reduced, delaying or avoiding its saturation and, hence, the uplift force.
- The drainage flow rate increased significantly with the inlet discharge flow when the drainage layer was not saturated; however, it (expressed as a fraction of the inlet flow) decreased with inlet flow if the drainage layer was saturated.
- It should be noted that in some cases, positive pressures, although low, were detected in the lower part of the riser. In these cases, the air vents presumably allowed the water to enter the drainage layer if air vents were located at the base of the riser, as was the case for the Armorwedge™ block.
- In the upper part of the channel, the hydrodynamic stabilizing force increased systematically with the discharge flow. The ACUÑA block was more stable than the Armorwedge™ block for all the tested cases. In the lower part of the channel, the stabilizing force was reduced with the discharge flow due to the saturation of the drainage layer and uplift pressures appearing at the base of the block. In this situation, the Armorwedge™ block was more stable than the ACUÑA block.
- When the joints between blocks were sealed, and the drainage layer was unsaturated, the stabilizing forces increased with the discharge flow, and the ACUÑA block was more stable than the Armorwedge™ block for all cases with the skimming flow regime.
- In both WSBs, the longitudinal joints between blocks were the source of the highest percentage of the total leakage flow. In addition, these leaks came mostly from the upper area of the flume.
- Although joint sealing is not a usual practice, it is advisable to consider the benefits and implement a cost-effective way for sealing the joints in new WSB dam protection against overtopping or spillways, especially in the upper sections.

## 5. Patents

The main result of the research is the development of a new design of WSB (Figure 26) termed ACUÑA (patented in Spain in May 2017 with the code ES2595852). The main difference of the ACUÑA model as compared with the previously developed WSBs is the position of the air vents located in the upper area of the riser. Additionally, the edge between the block tread and the riser was chamfered to facilitate the development of negative pressures in that area. This new layout reduces the leakage flow to the granular drainage layer. In addition, the ACUÑA block has a transverse orifice with the objective of transmitting the negative pressures to the lateral faces. This orifice presents other constructive advantages; it passes through the center of gravity to ease the transportation

to the construction site and also allows the tying of each row of blocks with a cable to avoid failure of an isolated block.

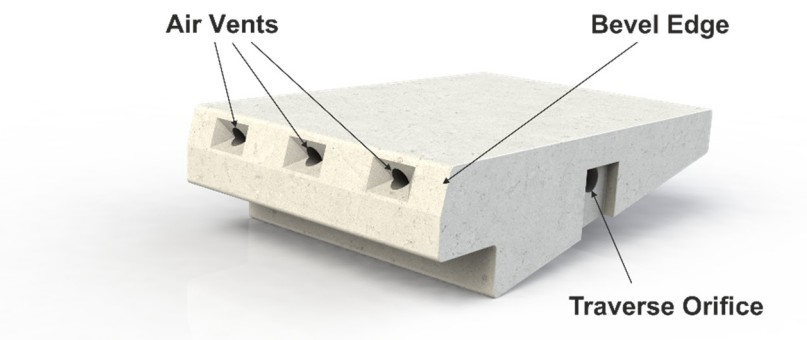

**Figure 26.** ACUÑA block (patent number ES2595852).

**Supplementary Materials:** The following are available online at https://zenodo.org/record/4957 363#.YMi660wRVPY, Figure S1: (**a**) Aerial view of Barriga dam. Burgos, Spain (Source: Regional Authority of Castilla y León, JCYL) (**b**) View from the top of the dam (**c**) Spillway dicharge (Source: José Manuel Ruiz, JCYL, May 2008) (**d**) Spillway (**e**) and (**f**) Details of the Amorwedge$^{TM}$ WSB chute spillway of the Barriga dam; Figure S2: Experimental setup. (**a**) and (**b**) Schemes. (**c**) Leakage flow collection system (**d**), Stilling basin, triangular thin-plate weir for leakage flow measurement and rectangular thin-plate weir for discharge flow measurement (**e**) ACUÑA WSBs placed over a fixed metallic grid (free drainage condition, *d*1) (**f**) ACUÑA WSBs placed over a layer of homogeneous gravel (granular drainage condition, *d*2); Figure S3: Inception point location; Figure S4: Uniform flow depth location; Figure S5: Mean pressures and standard deviation on the tread of Amorwedge$^{TM}$ block at different rows of the chute for skimming flow and unit flow: (**a**) 0.12 m$^2$ s$^{-1}$ (**b**) 0.16 m$^2$ s$^{-1}$ (**c**) 0.20 m$^2$ s$^{-1}$ (**d**) 0.24 m$^2$ s$^{-1}$; Figure S6: Average and standard deviation of the hydrodynamic pressures registered on the block riser of Amorwedge$^{TM}$ block at row 25 of the chute for a different unit discharges simulate free drainage conditions (*d*1): (**a**) Average results compared (**b**) 0.04 m$^2$ s$^{-1}$ (**c**) 0.08 m$^2$ s$^{-1}$ (**d**) 0.10 m$^2$ s$^{-1}$ (**e**) 0.12 m$^2$ s$^{-1}$ (**f**) 0.16 m$^2$ s$^{-1}$ (**g**) 0.20 m$^2$ s$^{-1}$; Figure S7: Average and standard deviation of the hydrodynamic pressures registered on the block riser of Amorwedge$^{TM}$ block at row 25 of the chute for a different unit discharges simulate the blocks laid over an impervious soil with an intermediate and permeable bedding granular layer (*d*2): (**a**) Average results compared (**b**) 0.04 m$^2$ s$^{-1}$ (**c**) 0.08 m$^2$ s$^{-1}$ (**d**) 0.10 m$^2$ s$^{-1}$ (**e**) 0.12 m$^2$ s$^{-1}$ (**f**) 0.16 m$^2$ s$^{-1}$ (**g**) 0.20 m$^2$ s$^{-1}$; Figure S8:Average and standard deviation of the hydrodynamic pressures registered on the block riser of Amorwedge$^{TM}$ block at row 10 of the chute for a different unit discharges simulate free drainage conditions (*d*1): (**a**) Average results compared (**b**) 0.04 m$^2$ s$^{-1}$ (**c**) 0.08 m$^2$ s$^{-1}$ (**d**) 0.10 m$^2$ s$^{-1}$ (**e**) 0.12 m$^2$ s$^{-1}$ (**f**) 0.16 m$^2$ s$^{-1}$ (**g**) 0.20 m$^2$ s$^{-1}$; Figure S9: Average and standard deviation of the hydrodynamic pressures registered on the block riser of Amorwedge$^{TM}$ block at row 10 of the chute for a different unit discharges simulate the blocks laid over an impervious soil with an intermediate and permeable bedding granular layer (*d*2): (**a**) Average results compared (**b**) 0.04 m$^2$ s$^{-1}$ (**c**) 0.08 m$^2$ s$^{-1}$ (**d**) 0.10 m$^2$ s$^{-1}$ (**e**) 0.12 m$^2$ s$^{-1}$ (**f**) 0.16 m$^2$ s$^{-1}$ (**g**) 0.20 m$^2$ s$^{-1}$; 00_Experimental_set_up (3 videos); 01_No_saturation_d2 (1 photo and 1 video); 02_Suction (2 videos); 03_w1_d1_2017 (1 video); 04_w2_d1_2017 (2 videos); 05_w1_d2_2018 (2 photos and 1 video); 06_w2_d2_2018 (2 videos); 07_w2_d2_2019_no_sealing (3 videos); and 08_w2_d2_2019_sealing_8rows (2 photos and 2 videos).

**Author Contributions:** Conceptualization, F.J.C., M.Á.T. and R.M.; methodology, F.J.C.; formal analysis, F.J.C., M.Á.T. and R.M.; investigation, F.J.C.; data curation, F.J.C.; visualization, F.J.C.; numerical modelling, J.S.M.; writing—original draft preparation, F.J.C.; writing—review and editing, R.M. and M.Á.T.; supervision, R.M. and M.Á.T. All authors have read and agreed to the published version of the manuscript.

**Funding:** This research was funded by the Spanish Ministry of Science and Innovation (Ministerio de Ciencia e Innovación, MICINN) through the projects ACUÑA (grant number IPT-2011-0997-020000), DIABLO (grant number RTC-2014-2081-5) and PABLO (grant number RTC-2017-6196-5). In addition, the ACUÑA project was partially financed by the European Regional Development Fund (ERDF) of the European Commission. Finally, the project InnovaDAM (grant number STARTUP RIS3-2019/L1-450), awarded to ACIS2in and funded by the Regional Authority of Madrid (CAM) and partially funded by the European Regional Development Fund (ERDF) of the European Commission, served to disseminate the technology and the works carried out in the previous projects to interested public and private entities in Spain during 2020.

**Institutional Review Board Statement:** Not applicable.

**Informed Consent Statement:** Not applicable.

**Data Availability Statement:** MDPI Research Data Policies.

**Acknowledgments:** The authors would like to acknowledge the collaboration of Luis Ruano[†] from PREHORQUISA, without whose involvement these projects would not have been possible, and the Red de Aulas CIMNE. The authors would also like to acknowledge Cristian Ponce-Farfán, Ricardo Monteiro-Alves, Javier Peraita and León Morera from the Dam Safety Research Group (SERPA) of UPM; Lucía Turrero, José Luis García-Ramos and Luis Balairón from CEDEX; Fernando Salazar from CIMNE; and Luis Jaime Caballero for the support to this research.

**Conflicts of Interest:** The authors declare no conflict of interest.

## Nomenclature

| | |
|---|---|
| $C_u$ | uniformity coefficient |
| $d1$ | free drainage condition |
| $d2$ | the granular layer drainage condition |
| $D_{10}$ | size of which 10% of the particles, in weight, are finer (m) |
| $D_{50}$ | size of which 50% of the particles, in weight, are finer (m) |
| $F^*$ | roughness Froude number |
| $h_c$ | critical depth |
| $h_s$ | height of the block riser |
| $L_i$ | longitudinal distance between the critical depth position on the crest of the dam and the horizontal face where the inception point is located |
| $L_u$ | longitudinal position from the dam or flume crest of the beginning of the quasi-uniform region |
| $l$ | the partially exposed length of the top surface of the block |
| $l_s$ | the total exposed length of the top surface of the block |
| $L'$ | partial length of the base of the block |
| $L'_b$ | the total length of the base of the block |
| $p$ | average hydrodynamic pressure |
| $q$ | flow-rate inlet in the chute |
| $q_d$ | drainage flow-rate |
| $w1$ | Armorwedge$^{TM}$ block |
| $w2$ | ACUÑA block |
| $x$ | coordinate horizontal to the crest of the dam |
| $y$ | coordinate perpendicular to the crest of the dam |
| $\gamma$ | water volumic weight |
| $\theta$ | the angle of the slope of the chute |

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
