# Peer review of "Hydrodynamic Performance and Design Evolution of Wedge-Shaped Blocks for Dam Protection against Overtopping"

_water, doi:10.3390/w13121665_

Round 1

Reviewer 1 Report

See the attached pdf file

Author Response

Dear reviewer.

The responses are in the attached file.

Kind regards.

Reviewer 2 Report

Overview:

The manuscript presents results from extensive experimental research relating to protection of embankment dams against potential overtopping. The topic is of importance for the field of dam safety and highly relevant for the special issue.

This reviewer recommends publication of this paper after a revision, as further listed below.

The review is organized as follows:

0              General 1

1              The Introduction and background. 1

2              Chapter 2, material and methods. 2

3              Chapter 3 Results and discussion. 2

4              Chapter 4 Conclusions. 3

5              Comments on Figures. 3

6              Comments on Tables. 4

0            General

  1. The English language needs extensive editing. Please ask an English native speaking person to read through and edit the language.
  2.  
  3. The text is too small in many of the figures. Please go systematically through all the figures and make sure that the font sizes fulfil the requirement of the journal.

1            Chapter 1 Introduction and background

The introduction provides a good overview of the field, below are some points to consider.

  1. Please go critically through the introductory section aiming at improving the flow of the text and the information, as well as the English language. Avoid very short paragraphs such as in line 49 to 51, and line 60 to 61.
  2. Please make the goal of the study clearer in the introduction, for example it is more clearly stated later in lines 143 to 152.
  3. Consider moving Fig. 7 into the introduction as the first figure. This will give a nice presentation from the start of the use of the WSBs in a real structure.
  4. Consider moving sentence in line 60 to 61, for example to the end of the first paragraph (line 37).
  5. In line 33 you say that the modular elements are made of “mass concrete”. Mass concrete is usually not high strength concrete as it usually is of great volume. Recommend removing the word “mass”.
  6. In the sentence ending with the word “block” in line 89, please add reference to Armorwedge the block that you are improving; for example (starting in line 88): “..improve the behavior of pre-existing Armorwedge blocks. “
  7.  

2            Chapter 2, material and methods

  1. Please go critically through the section aiming at improving the flow of the text and the information, as well as the English language. As before, avoid very short
  2. paragraphs such as for example in line 140, and 141 to 142.
  3. Consider adding a subchapter 2.1 Experimental setup, starting in line 93 and ending in line 140.
  4. Line 117-118, after all the rows add “ (coloured black in Fig. 2)”. (“…the rows 5, 10, 15, 25, 30 and 35 (coloured black in Fig. 2) )”.
  5. Please make sure that Fig.2 a, b and c are on the same page.
  6. Consider adding a subchapter 2.3 Experimental testing program, starting in line 141 (but relocate the first sentence) and ending in line 209.
  7. In the suggested new subchapter on the experimental testing program, please add a table listing all the tests that you carried out, so that the (impressive) extent of the work can be clearly understood.
  8. In line 142, add a reference to Table 1. Also move the sentence in line 141 to 142, to a a more fitting position within the suggested new subchapter on the testing program.
  9. In line 155, you mention “following phases of the test program” and refer to Figure 4b, that does not show these phases. Please define the phases of the test program.

3            Chapter 3 Results and discussion

  1. Please go critically through the section aiming at improving the flow of the text and the information, as well as the English language. As before, avoid very short paragraphs.
  2. Consider moving lines 211 to 223 (including the figure 9) to the conclusion section. That is, make this the outcome of the result. This could for example be the last paragraph of the article.
  3. The subchapter 3.1 seems to be misplaced. This reviewer understands the information in this chapter as a part of the planning phase for the experiments, and also to some extent a theoretical background. Consider moving this subchapter into the Chapter 2, before the previously suggested subsection 2.2 (which then becomes subsection 2.3), and make clear that this is a selection of flow regimes for the tests.
  4. Please explain/define the critical depth, hc. Also please show this on figure 9.
  5. Please repeat the definition tread length, etc, given in lines 283 to 284 to join the text close to Figure 10.
  6. This reviewer understands in so that the results from the tests itself is first introduced in the current section 3.2. The authors present an extensive amount of results. Please provide an overview of the content of the results chapter in the beginning of the chapter.
  7. In line 332, add a reference to figure 16.
  8. Note that sometimes you refer to the blocks as w1 and w2, and sometimes by name. Please, make this consistent in the paper.
  9. Line 369, make clear what the proposed block is (refer to Acuna).
  10. Shouldn’t 3.2.4 Hydrodynamic forces on the WSBs, be subchapter 3.3?
  11. The authors have chosen to combine the result and discussion. Generally this is in two separate section, however, in this case this seem to fit well together. Particularly, if you reorganize the text for a better flow, e.g. as suggested above.
  12.  

4            Chapter 4 Conclusions

  1. Please go critically through the section aiming at improving the English language.
  2. When you list the main conclusions, please indicate to which block you are referring Armorwedge or Acuna, otherwise the conclusions are somewhat confusing.
  3. As previously mentioned: Consider moving lines 211 to 223 (including the figure 9) to the conclusion section. That is, make this the outcome of the result. This could for example be the last paragraph of the article.

5            Comments on Figures

Figure 2: Please keep the figure on the same page.

Figure 4: Please indicate the units of the dimensions in the figure. The dimensions are given as for example 243.3, 27.3 etc., Assuming that these are mm, this is very accurate numbers. Should it not be 243 and 27 mm. Please make sure that the accuracy of the dimension is appropriate.

Figure 5: note that (b) is misaligned. (Refer to d1 and d2 in the caption).

Figure 7: Consider making fig. 7 the first figure in the manuscript.

Figure 9: It would be clear if you could, in the figure, point to the features you mention in line 215 to 220.

Figure 10: Please indicate hc in this figure. Also the letters, h, l etc in the figure should be in italic font, same as in the text.

Figure 11, 12, 13: Please consider enlarging the font size of the axis title for readability.

Figure 14: Please consider enlarging the font size of the axis title for readability. In the caption provide information on the drainage condition.

Figure 15 to 20: Please consider enlarging the font size of the axis title for readability. Please add a line where the pressure is zero.

Figure 20: Note spelling of longitudinal sealing.

Figure 22 and 25: Note the caption should give info on a) to g). Also, please consider enlarging the font size of the axis title for readability.

Figure 26: Note that the discharge information (i to ix) within each subfigure is so small that it is almost invisible, please enlarge this. Also, please consider enlarging the font size of the axis title for readability.

Figure 28: Please consider adding information on the discharge into the figure. Note also spelling fo hydrodynamic in the vertical axis title.

6            Comments on Tables

Table 2: Please consider removing the decimal point, e.g. 779,1 becomes 779; 683,6 becomes 684, etc.

Table 3: Why do you not always calculate for the same row?

Add a table with overview of all the test conducted.

Author Response

(The authors gave the same response as above.)

Reviewer 3 Report

Dear authors,

The research topic that you presented in this paper focusing on hydrodynamic performance and design evolution of wedge-shaped blocks is very interesting. Please find below a few remarks reflecting on your research that I believe would add to the overall quality of the manuscript.

General remarks:

Manuscript title is a bit misleading – it suggests that WSB are performing an action, evolving and protecting dam against overtopping. I suggest that you add more keywords for clarification, along the following train of thought: “hydrodynamic performance”, “design evolution”. Also, the WSB are primarily protecting the spillway from erosion, and indirectly dam from overtopping (probably not in significant way, but since overtopping hazard it is not quantified, I wouldn’t even open this topic). Therefore, I suggest to rephrase the title in general.

Entire manuscript is divided into small paragraphs, some containing no more than one sentence. Therefore, the flow of the section is disrupted. Please combine the paragraphs into less larger ones that follow the same train of thought.

Introduction and background section the paper focuses entirely on the introduction and state of the art review is missing. Please add one paragraph describing relevant papers that have published similar research.

Materials and Methods section is doo descriptive – it is lacking the quantification of the methodology used for obtaining and analyzing the results. Entire paper is lacking Methodology section. Some of the methodology is blended within the results but just referenced without listing how was it adapted for specific application in this research.

In line with previous remark, Figure 12. is redundant as the depicted data is presented in the Table 2, and reference data from other authors can be added to the same table.

The same is valid for Figure 13 and Table 3.

Since the paper is already very extensive, I suggest that results and discussion are divided into two separate sections for clarity.

Results for pressures on the block tread are not comparable between Amorwedge and ACUÑA blocks. I suggest that you move the particular results (i.e. Figures 14 – 17) to Annex and keep only com parable results in the main paper body (i.e. Figures 19 – 20).

Results for pressures on the block riser are given as a time series, but authors haven’t explained why this is relevant or at which moment is the flow fully developed during the experiment, especially as preceding analyses were done for stationary or averaged conditions.

In line with the previous remark, it is not clear if the results depicted on Figure 21 show increasing flow rates as presented in Figures 26 – 28.

It is not clear if the stationary conditions for each flow rate are achieved during the experiments – in the section 3.2.4. these results are used for establishing correlation of with hc hs^-1 but there is no explanation how did the authors check if hydrodynamic environment has stabilized in order to finalize the experiment.

Overall, discussion mostly consists of readout of the measured data, without proper discussion and relevance of the obtained results.

Specific remarks:

Some of the keywords are not mentioned in the abstract – “ACUÑA”, „embankment dam”, “precast concrete”. Especially ACUÑA patent needs to be included in the abstract since it is the central point of the paper. I suggest to delete the keywords “embankment dam” and “precast concrete” as they are too general.

On line 450 there is a typo – “cero”.

Author Response

(The authors gave the same response as above.)

Round 2

Reviewer 1 Report

I thank the authors for their effort to answer my doubts and requests for changes. Many of them have been done and, in my opinion, the article is now more concise and readable, while maintaining all the most relevant aspects of the experiment in the main text.

I still have only a few minor aspects, essentially correcting the text, which I submit to the attention of the authors for correction.

Line 157 - “Figure 10” is now Figure 5

Line 158 - "l" in italics

Lines 158-159 and lines 294-295 - this lines can be omitted, as these definitions have already been given at lines 152-155

Lines 186-187 - Rewrite the sentence better

Line 236 - D50 the subscript must not be in italics

Lines 280-281-283 w1, w2, d1, d1 in italics

Line 383 d2 in italics

Line 576 "Amorwegde" is Armorwedge

Author Response

The authors thank for their suggestions, which have undoubtedly resulted in a clear improvement in the quality of the article. Find attached the response to the last revison.

Reviewer 3 Report

Dear authors,

Thank you for the update of the manuscript according to my remarks. I feel that paper is suitable for publishing in present form. 

Author Response

The authors thank for their suggestions, which have undoubtedly resulted in a clear improvement in the quality of the article.